# A combined adjuvant approach primes robust germinal center responses and humoral immunity in non-human primates

Adjuvants and antigen delivery kinetics can profoundly influence B cell responses and should be critically considered in rational vaccine design, particularly for difficult neutralizing antibody targets such as human immunodeficiency virus (HIV). Antigen kinetics can change depending on the delivery method. To promote extended immunogen bioavailability and to present antigen in a multivalent form, native-HIV Env trimers are modified with short phosphoserine peptide linkers that promote tight binding to aluminum hydroxide (pSer:alum). Here we explore the use of a combined adjuvant approach that incorporates pSer:alum-mediated antigen delivery with potent adjuvants (SMNP, 3M-052) in an extensive head-to-head comparison study with conventional alum to assess germinal center (GC) and humoral immune responses. Priming with pSer:alum plus SMNP induces additive effects that enhance the magnitude and persistence of GCs, which correlate with better GC-$T_{FH}$ cell help. Autologous HIV-neutralizing antibody titers are improved in SMNP-immunized animals after two immunizations. Over 9 months after priming immunization of pSer:alum with either SMNP or 3M-052, robust Env-specific bone marrow plasma cells (BM $B_{PC}$) are observed. Furthermore, pSer-modification of Env trimer reduce targeting towards immunodominant non-neutralizing epitopes. The study shows that a combined adjuvant approach can augment humoral immunity by modulating immunodominance and shows promise for clinical translation.

Most licensed vaccines prevent disease by inducing long-lasting, protective antibody responses[1]. Broadly neutralizing antibodies (bnAbs) isolated from human immunodeficiency virus (HIV)-infected individuals are capable of protecting non-human primates from virulent SHIV infection, and those outcomes have guided extensive research efforts with the hope of developing an antibody-based HIV vaccine[2].

B cells secreting affinity-matured antibodies are the output of germinal centers (GC). GCs are microanatomical sites of B cell clonal expansion and selection—each GC reaction evolution in miniature—driven by competition for antigen and GC T follicular helper (GC-$T_{FH}$) cell help[3,4]. With each selection cycle within a GC, B cells have the potential to gain in antibody affinity, fine-tuning recognition of their

cognate antigen, before eventual differentiation and exit as either a plasma cell ($B_{PC}$) that constitutively secrete antibodies or a resting memory B cell ($B_{Mem}$) capable of participating in recall responses upon pathogen re-exposure[3,4]. In the context of vaccines, adjuvants help drive these immunological processes. Adjuvants can function by a wide variety of mechanisms of action, with different consequences for different components of the innate and adaptive immune systems[5,6]. Adjuvants can play critical roles in enhancing the magnitude and durability of an immune response. Aluminum hydroxide (alum) is the most prevalent adjuvant used in licensed human vaccines. Although alum often induces less potent immune responses compared to other adjuvants in preclinical and clinical studies[7,8], it continues to be a "gold

✉e-mail: djirvine@mit.edu; shane@lji.org

standard" for almost all new adjuvant development. Saponin formulations have also been extensively studied as vaccine adjuvants. One form of a saponin of interest is immune-stimulating complexes (ISCOMs), consisting of saponin, phospholipid, and cholesterol mixtures that self-assemble into nanoparticles. Novavax's protein subunit vaccine (NVX-CoV2373) for severe acute respiratory syndrome coronavirus 2 (SARS-CoV-2) is formulated with an ISCOMs-type saponin adjuvant, Matrix M, that has shown high efficacy in phase 3 clinical trials and is now approved for use and administered globally[9]. ISCOMs incorporating the TLR4 agonist MPLA, termed saponin/MPLA nanoparticles (SMNP), is a newly developed adjuvant found to enhance lymphatic trafficking of antigen and promote robust germinal center B cell (B$_{GC}$) responses[9].

While B cells theoretically can target a wide continuum of epitopes on a protein antigen surface, antibody responses in reality often target a select number of sites, due to a phenomenon termed B cell immunodominance[10,11]. Immunodominant but non-neutralizing epitopes on recombinant HIV envelope trimers (Env) include the neoantigen exposed base of the soluble recombinant trimer and trimer-breakdown products from in vivo degradation ("dark antigen")[12–14]. Evidence of immunodominance impairing neutralizing antibody (nAb) development is observed in the varied levels of tier-2 nAb induction in native-like trimer-immunized rhesus monkeys (RMs), where BG505 SOSIP Env-binding IgG titers were not predictive of autologous BG505 nAb titers[15]. Strategies to modulate B cell immunodominance will likely be key to inform rational vaccine designs, particularly for difficult pathogens[10,11,16]. Modifying antigen characteristics ("intrinsic properties") – antigen affinity and precursor frequency[17], antigen valency[18], and quantity of T$_{FH}$ help[19,20] –can profoundly impact B cell competitive fitness and composition in the GCs.

In addition to such intrinsic properties of a vaccine, changes to immunogen kinetics ("extrinsic properties") prolonging antigen availability by slow delivery can increase the magnitude of the GC response[12], prime long-lasting GCs[21], and alter the antibody repertoire, allowing for a larger diversity in the epitopes targeted. These and other attributes may also be impacted by the use of different adjuvants, but these characteristics are currently not well understood. Phosphoserine-tagging of an immunogen (pSer) mixed with alum (pSer:alum) is a promising approach that alters multiple facets of vaccine delivery: (i) it slows antigen clearance kinetics for sustained antigen bioavailability, (ii) provides increased avidity from antigen-coated alum particle uptake, and (iii) promotes epitope shielding by directed orientation of antigen on the alum surface to shield off-target epitopes such as the base of the trimer[22,23]. However, alum remains an adjuvant that promotes modest inflammation in draining lymph nodes compared to other molecular adjuvants such as TLR agonists and saponins, and thus we hypothesized that there could be value in combining alum:pSer delivery with additional co-adjuvants. As pSer:alum delivery potentially impacts multiple factors involved in B cell immunodominance and GC biology, it is unclear what the range of impact the pSer modification may have on adaptive immunity in the context of additional adjuvants with different mechanisms of action. Is there a synergistic effect in a combined approach of pSer with multiple adjuvants, such as pSer:alum with SMNP? How does modifying multiple factors at once change their interplay or interdependence on each other? How do these adjuvants compare head-to-head and to the gold standard, conventional alum? Here, we immunized RMs with different adjuvants in an effort to answer these questions.

## Results

To characterize the immunogenicity of different adjuvants in non-human primates (NHPs) as a model of human biology, five groups of RMs ($n = 6$/group) were subcutaneously immunized with a recombinant native-like HIV Env trimer[24], MD39 (50 μg per side), at week 0 and subsequently boosted at week 10, week 24, and week 40. Five adjuvant

technologies were tested: (i) aluminum hydroxide (MD39 adsorbed to alum, Alhydrogel), (ii) pSer:alum (pSer-MD39 bound to alum), (iii) pSer:alum + SMNP (pSer-MD39 bound to alum mixed with SMNP), (iv) SMNP (soluble MD39 mixed with SMNP), and (v) pSer:alum-3M-052 (pSer-MD39 trimer bound to alum bearing preadsorbed 3M-052 TLR7/8 agonist). We have previously described an approach to augment protein subunit vaccines by modifying the extrinsic properties of a vaccine–antigen delivery kinetics. Site-specific introduction of pSer tags onto protein immunogens allows for a tight binding of the antigen to the surface of alum via a ligand exchange reaction. This approach has been applied with various immunogens, from small protein constructs such as HIV engineered outer domain gp120 containing the CD4 binding site (eOD-GT8) and SARS-CoV-2 receptor binding domain (RBD) to larger HIV Env trimers (MD39)[22,23]. pSer-conjugated MD39 exhibited high levels of initial binding to alum and retention on alum in the presence of mouse and RM serum, and preserved antigenicity when bound to alum (Supplementary Fig. 1a–c). All animals were immunized bilaterally in the left and right thigh. As germinal center (GC) responses in contralateral limbs were found to be largely independent after a priming immunization[15,25,26], bilateral immunizations increase the number of lymph nodes (LNs) available for sampling GCs, thus increasing statistical power of NHP immunization priming studies. LN fine needle aspirates (FNAs) were performed serially to sample GCs in draining inguinal LNs (iLNs) (Fig. 1a). Unmodified MD39 adsorbed to alum was used as a classical adjuvant approach (Group 1). Alum plus pSer-modified MD39 was a direct test of the immunogenicity of a pSer-modified protein (Group 1 vs 2). Group 3 represented a first test of the adjuvanticity of a combination of pSer and SMNP technologies in a large animal model. A previous study done in mice investigating humoral responses to immunization combining pSer-RBD with an alum-binding co-adjuvant, SMNP (pSer-RBD:alum + SMNP), resulted in longer antigen retention, higher anti-RBD IgG titers, and enhanced neutralizing antibody responses over pSer-RBD:alum, indicating synergy between co-adjuvants[23]. SMNP plus soluble Env trimer was Group 4, providing a comparison of SMNP versus conventional alum ("alum", Group 1 vs 4) as well as SMNP with and without pSer:alum (Group 3 vs 4). A previous NHP study utilized a total dose of 750 μg of SMNP (375 μg administered at two sites bilaterally) and was well tolerated[9]. Animals in Group 3 and 4 of this study received a lower total dose of 375 μg of SMNP (187.5 μg administered bilaterally), comprised of a 10:1 molar ratio of saponin Quil-A and MPLA. The amount of MPLA used is 10-times less than in AS01$_B$, a licensed liposomal saponin-based adjuvant used in the Shingrix vaccine[9]. 3M-052, a synthetic TLR7/8 agonist adsorbed onto alum[27], was used in the 5$^{th}$ animal group, also combined with pSer technology, to assess immunogenicity in comparison to alum (Group 1) and pSer:alum + SMNP (Group 3). It was previously reported that 75 to 750 μg doses of 3M-052 encapsulated in PLGA nanoparticles induced robust autologous tier-2 neutralizing antibodies after 3-4 prime-boost Env trimer immunizations of RMs, and induced durable bone marrow-resident long-lived plasma cells[28,29]. Here, animals in the 3M-052 adjuvant group received a 5 μg 3M-052 dose formulated with alum, reflective of the dose and formulation of 3M-052 that was found to be safe and tolerable in an ongoing human vaccine clinical trial (NCT04177355).

### pSer:alum-mediated antigen delivery work synergistically with SMNP adjuvant to prime larger germinal center responses

After a single priming immunization, pSer:alum + SMNP induced higher total GC B cell (B$_{GC}$, CD38$^-$CD71$^+$) frequencies than all other immunization groups, peaking at week 3 (Fig. 1b–d, Supplementary Fig. 2a, b). Median peak B$_{GC}$ cell frequency observed post-prime for pSer:alum + SMNP was approximately 5.6-times higher than alum ($P = 0.0012$), 5.1-times higher than pSer:alum (P < 0.0001), and 3.8-times higher than SMNP ($P = 0.046$). The median peak B$_{GC}$ cell frequency for pSer:alum-3M-052 was 2.3-times higher than pSer:alum

($P = 0.032$). When compared to pSer:alum-3M-052, pSer:alum + SMNP induced 2.2-times higher $B_{GC}$ cell response ($P = 0.034$) at their respective median peaks (Supplementary Fig. 2c).

Analysis of the $B_{GC}$ cell kinetics post-prime revealed that adjuvanting with pSer:alum + SMNP induced a much larger cumulative $B_{GC}$ cell response compared to alum (area under the curve [AUC], $P = 0.0009$) or pSer:alum ($P = 0.0014$, Fig. 1c, d). The combination of SMNP plus pSer:alum led to a significantly larger cumulative $B_{GC}$ cell response than pSer:alum ($P = 0.0014$) and trended higher than SMNP Group 4 ($P = 0.18$, Fig. 1c, d), which may suggest that the two adjuvant technologies cooperate to elicit better priming of B cells and recruitment into GCs. Modest $B_{GC}$ cell frequencies were observed post-prime in Group 1 and Group 2, underscoring the importance of potent adjuvants to prime large GC responses to a challenging antigen such as fully glycosylated Env trimer.

The GC responses were tracked serially after a 1st booster immunization at week 10 and a 2nd booster immunization at week 24 (Fig. 1a). After the 1st booster immunization, pSer:alum + SMNP continued to elicit robust $B_{GC}$ cell responses compared to alum alone ($P = 0.0004$)

and pSer:alum ($P = 0.045$, Supplementary Fig. 2d). SMNP alone also induced significantly higher $B_{GC}$ cell frequencies compared to alum alone ($P = 0.029$, Supplementary Fig. 2d). pSer:alum-3M-052 immunization maintained a durable GC response with mean $B_{GC}$ cell frequencies between 7.6 to 11% throughout the course of the study (Fig. 1c, d, Supplementary Fig. 2d, e). After the 2nd booster immunization at week 24, there was no significant difference in the $B_{GC}$ cell activity between each group (Supplementary Fig. 2e).

Env-binding $B_{GC}$ cells (CD38⁻CD71⁺/Env⁺/⁺, Figs. 1b, e, f, Supplementary Fig. 2f–i) and total Env-binding B cells (CD3⁻CD20⁺/Env⁺/⁺, Supplementary Fig. 2j–m) were quantified. Following priming immunization, pSer:alum + SMNP elicited a robust, early Env-binding $B_{GC}$ cell response compared to each group: 10-times higher than alum alone ($P = 0.0005$), 6.2-times higher than pSer:alum ($P = 0.0007$), and 4.8-times higher than SMNP alone ($P = 0.0076$). pSer:alum + SMNP also induced a Env-binding $B_{GC}$ cell response 3.0-times stronger than pSer:alum-3M-052, of borderline statistical significance ($P = 0.055$). There was no significant difference in Env-binding $B_{GC}$ cell frequencies between pSer:alum and alum alone ($P = 0.55$, Fig. 1f). Interestingly,

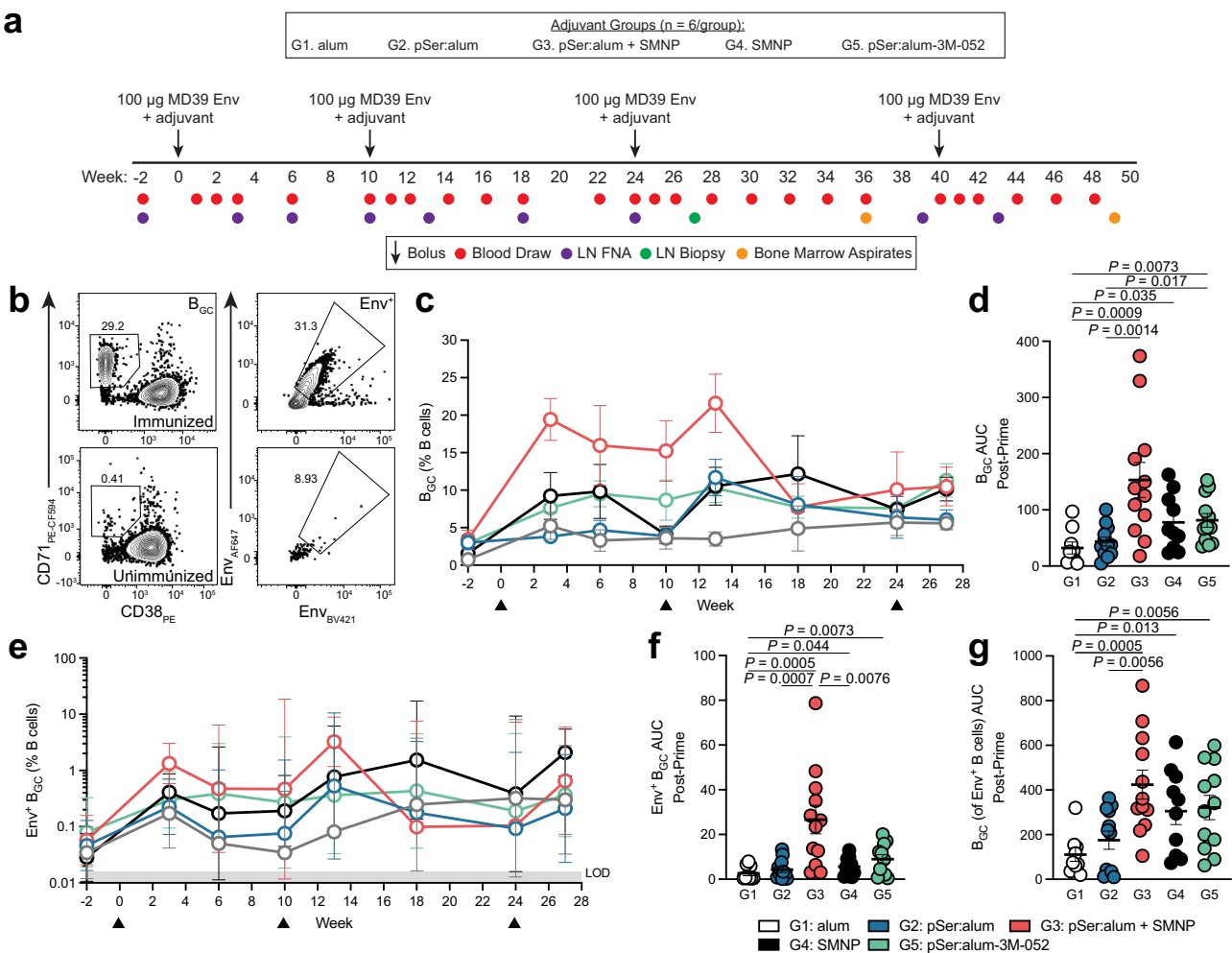

**Fig. 1 | pSer:alum modification and SMNP adjuvant work synergistically to promote germinal center activity. a** Schematic of the study where NHPs (n = 6 per group) were immunized with soluble native-like HIV BG505 Env trimer MD39 ("Env") with the different adjuvants at the indicated timepoints. **b** Flow cytometry gating of $B_{GC}$ cells (CD38⁻CD71⁺) and of Env-binding $B_{GC}$ (Env-AF647⁺Env-BV421⁺). **c** $B_{GC}$ cell frequency (CD38⁻CD71⁺) as a percentage of B cells. **d** Area under the curve (AUC) of $B_{GC}$ cell frequency as a percentage of B cells post-priming immunization. **e** Env-binding $B_{GC}$ frequency as a percentage of B cells. **f**, AUC of Env-binding $B_{GC}$ cell frequency post-priming immunization. **g** AUC of the proportion of Env-binding

B cell frequency that are $B_{GC}$ cells post-priming immunization. Black triangles represent time of immunization. Mean and SEM or geometric mean and geometric SD are plotted depending on the scale in all figures unless otherwise stated. For Fig. 1c-g, n = 12 samples (left and right LN, 6 animals per group). Statistical significance was tested using either unpaired two-tailed Mann-Whitney tests or Kruskal-Wallis test with Dunn's multiple comparisons test, depending on the objectives of the study. *$p < 0.05$, **$p < 0.01$, ***$p < 0.001$, ****$p < 0.0001$. Source data are provided as a Source Data File.

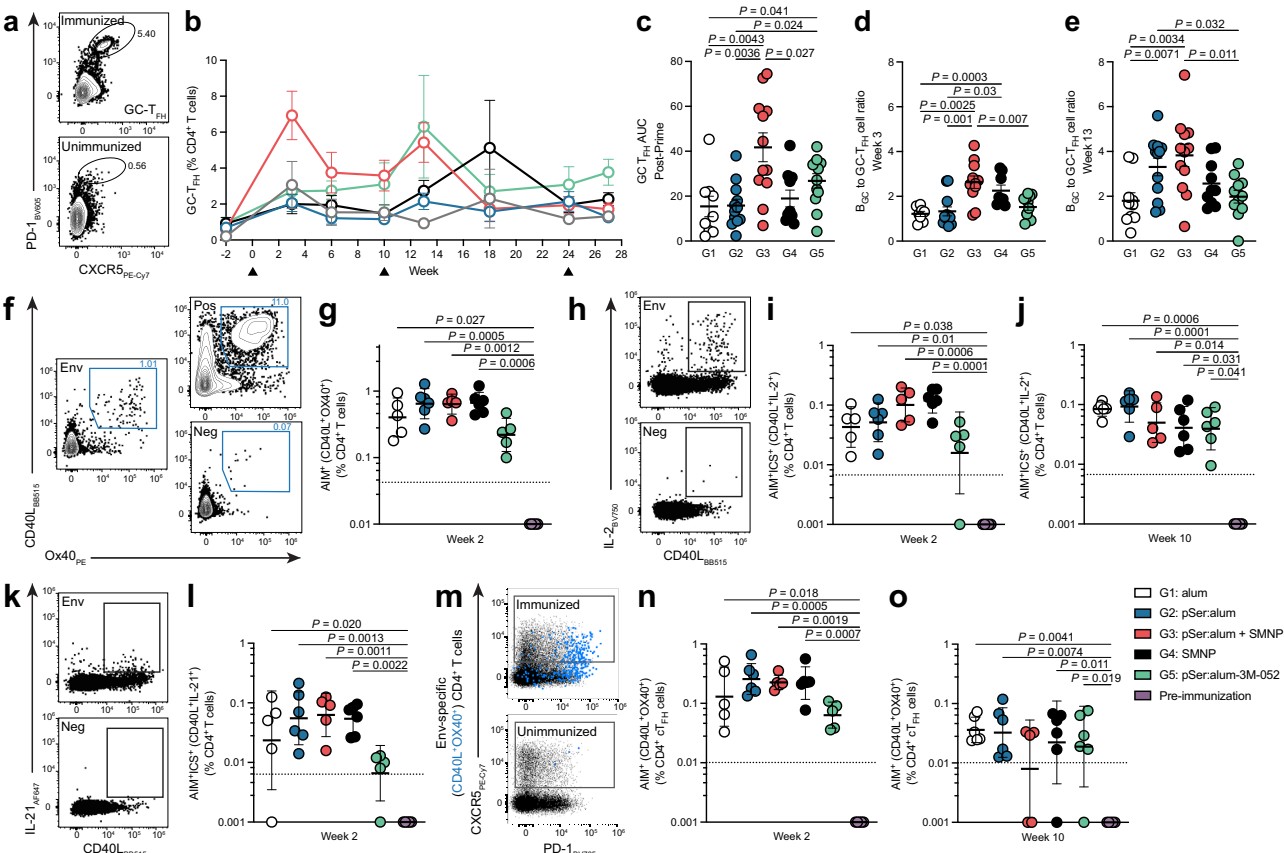

**Fig. 2 | Robust GC-T$_{FH}$ cell help and Env-specific T cell responses detected after priming immunization. a** Flow cytometry gating of GC-T$_{FH}$ cells (PD-1$^{hi}$CXCR5$^+$). **b** Longitudinal analysis of GC-T$_{FH}$ cell frequency as a percentage of CD4$^+$ T cells. **c** AUC of GC-T$_{FH}$ cells post-priming immunization. **d** B$_{GC}$ to GC-T$_{FH}$ cell ratio at week 3. **e** B$_{GC}$ to GC-T$_{FH}$ cell ratio at week 13. **f** Representative flow plot of an AIM assay performed at week 2 to detect Env-specific CD4+ T cells. Stimulation conditions include: negative, unstimulated DMSO control ("Neg"), 15-mer overlapping MD39-Env peptide megapool ("Env"), and positive control with superantigen staphylococcal enterotoxin ("Pos"). AIM assays were also performed at week 10. **g** AIM$^+$ (CD40L$^+$OX40$^+$) T cell frequency as a percentage of CD4$^+$ T cells at week 2. **h** Representative flow plots of an intracellular cytokine assay (ICS) to assess cytokine function in Env-specific CD4$^+$ T cells. CD40L$^+$IL-2$^+$ is shown with corresponding Env-stimulated sample and unstimulated control sample. **i,** AIM$^+$ICS$^+$ (CD40L$^+$IL-2$^+$) T cells as a percentage of CD4$^+$ T cells at week 2. **j** AIM$^+$ICS$^+$ (CD40L$^+$IL-2$^+$) T cells as a percentage of CD4$^+$ T cells at week 10. **k** Representative flow plots of an intracellular cytokine assay (ICS) to assess cytokine function in Env-specific CD4$^+$ T cells.

CD40L$^+$IL-21$^+$ is shown with corresponding Env-stimulated sample and unstimulated control sample. **l** AIM$^+$ICS$^+$ (CD40L$^+$IL-21$^+$) T cells as a percentage of CD4$^+$ T cells at week 2. **m** Representative flow plot of an AIM assay to detect Env-specific circulating T$_{FH}$ (cT$_{FH}$) cells. **n** AIM$^+$ (CD40L$^+$OX40$^+$) cT$_{FH}$ frequency as a percentage of CD4$^+$ T cells at week 2. **o**, AIM$^+$ (CD40L$^+$OX40$^+$) cT$_{FH}$ frequency as a percentage of CD4$^+$ T cells at week 10. Black dotted lines indicate the limit of quantification. Black triangles represent time of immunization. Mean and SEM or geometric mean and geometric SD are plotted depending on the scale in all figures unless otherwise stated. For Fig. 2b–e, $n$ = 12 samples (left and right LN, 6 animals per group). For Fig. 2f–o, $n$ = 6 per group. Statistical significance for GC-T$_{FH}$ cell frequencies (a-e) was tested using unpaired two-tailed Mann-Whitney test or Kruskal-Wallis test with Dunn's multiple comparisons test, depending on the objectives of the study. Statistical significance for Env-specific AIM assays (f-o) were tested using Kruskal-Wallis test with uncorrected Dunn's multiple comparisons test. *$p$ < 0.05, **$p$ < 0.01, ***$p$ < 0.001, ****$p$ < 0.0001. Source data are provided as a Source Data File.

immunization with pSer:alum + SMNP elicited a larger proportion of Env-specific B cells that were B$_{GC}$ cells compared to pSer:alum ($P$ = 0.0056) and alum ($P$ = 0.0005), observed particularly post-priming immunization (Fig. 1g, Supplementary Fig. 2j–p). Similarly, SMNP induced a larger population of Env-specific B cells that were B$_{GC}$ cells compared to alum alone ($P$ = 0.013, Fig. 1g). The shift towards Env-binding B$_{GC}$ cells from total B cells seen in pSer:alum + SMNP, but not in pSer:alum, may suggest SMNP preferentially biases B cells towards B$_{GC}$ cell differentiation.

After the 1$^{st}$ booster immunization, Env-binding B$_{GC}$ cell frequencies in the pSer:alum + SMNP immunization group were again higher than alum ($P$ = 0.0034), pSer:alum ($P$ = 0.045), and pSer:alum-3M-052 ($P$ = 0.0204, Supplementary Fig. 2h) groups. After the 2$^{nd}$ booster immunization, there was no significant difference in Env-binding B$_{GC}$ cells (Supplementary Fig. 2i). Overall, these data demonstrate that pSer:alum + SMNP primed much more robust antigen-specific B$_{GC}$ cell responses than a conventional alum immunization,

and the most Env-binding B$_{GC}$ cells of any adjuvant approach tested here.

## Robust GC-T$_{FH}$ cell help and Env-specific T cell responses detected after priming immunization

B$_{GC}$ cells depend heavily on T$_{FH}$ cell help for initial B$_{GC}$ cell differentiation and subsequent somatic hypermutation (SHM) and antibody affinity maturation, as well as long-term humoral immunity[4]. A single bolus priming immunization of pSer:alum + SMNP with Env trimer elicited the largest GC-T$_{FH}$ cell response (PD-1$^{hi}$CXCR5$^+$ of CD4$^+$CD8a$^-$) of all immunization groups, peaking at week 3 (Fig. 2a–c, Supplementary Fig. 2a, Supplementary Fig. 3a). Post-prime GC-T$_{FH}$ cell frequencies after pSer:alum + SMNP were 2.7-times greater than alum (P = 0.0043), 2.6-times greater than pSer:alum ($P$ = 0.0036), and 2.2-times greater than SMNP ($P$ = 0.027, Fig. 2c). pSer:alum + SMNP continued to induce strong GC-T$_{FH}$ cell responses post-1$^{st}$ booster immunization, along with pSer:alum-3M-052 (Supplementary Fig. 3b). By the

$2^{nd}$ booster immunization at week 24, there were no significant differences between the groups (Supplementary Fig. 3c), highlighting that immunization with potent adjuvants may be most beneficial early in the immune response.

A metric for GC-$T_{FH}$ help quality is the ratio of $B_{GC}$ cells supported per GC-$T_{FH}$ cell[15]. At week 3, 3-weeks post-priming immunization, pSer:alum + SMNP immunization had induced $B_{GC}$:GC-$T_{FH}$ cell ratios 2.2-times greater than alum ($P = 0.0025$), 2.0-times greater than pSer:alum ($P = 0.001$), and 1.7-times greater than pSer:alum-3M-052 ($P = 0.007$). SMNP also induced higher quality GC-$T_{FH}$ cell help compared to alum ($P = 0.0003$) and pSer:alum ($P = 0.03$, Fig. 2d). At week 13, 3-weeks post-$1^{st}$ booster immunization, pSer:alum + SMNP continued to induce improved GC-$T_{FH}$ cell help compared to alum ($P = 0.0034$) and pSer:alum-3M-052 ($P = 0.011$). Interestingly, the $B_{GC}$:GC-$T_{FH}$ cell ratio for pSer:alum was between 1.7–1.8-times higher compared to alum ($P = 0.0071$) and pSer:alum-3M-052 ($P = 0.032$, Fig. 2e). The spike in total GC-$T_{FH}$ cell frequencies and the increased $B_{GC}$:GC-$T_{FH}$ cell ratios observed post-priming immunization imply that pSer:alum + SMNP promotes robust GCs both in the magnitude of GC-$T_{FH}$ cells and in the quality of GC-$T_{FH}$ cell help.

To identify Env-specific T cells, cytokine-independent activation-induced marker (AIM) assays were performed on peripheral blood mononuclear cells (PBMCs) 2-weeks and 10-weeks post-priming immunization (Fig. 2f–o, Supplementary Fig. 3d). At 2-weeks post-priming immunization, robust Env-specific (AIM$^+$, CD40L$^+$OX40$^+$) CD4$^+$ T cell responses were seen with alum alone (Kruskal-Wallis $P = 0.027$), pSer:alum ($P = 0.0005$), pSer:alum + SMNP ($P = 0.0012$), and SMNP alone ($P = 0.0006$) compared to the samples at the pre-immunization timepoint (Fig. 2g, Supplementary Fig. 3e). Only pSer:alum-3M-052 did not induce a significantly higher AIM$^+$ CD4$^+$ T cell response at week 2 compared to pre-immunization ($P = 0.24$, Fig. 2g); but by week 10, the AIM$^+$ CD4$^+$ T cell responses in the pSer:alum-3M-052 animals were significant (Supplementary Fig. 3e, f). Other AIM assay markers gave comparable results (Supplementary Fig. 3g–j). Intracellular cytokine analysis of Env-specific CD4$^+$ T cells (AIM$^+$ICS$^+$) at week 2 revealed that animals immunized with alum ($P = 0.038, 0.024, 0.039, 0.043$), pSer:alum ($P = 0.01, 0.015, 0.039, <0.0001$), pSer:alum + SMNP ($P = 0.0006, 0.0027, 0.0002, 0.0051$), and SMNP ($P = 0.0001, 0.008, 0.0002, 0.0004$) produced significantly higher amounts of IL-2, IFNγ, TNFα, and granzyme B, respectively, compared to pre-immune samples (Fig. 2h–i, Supplementary Fig. 3d, l, n, p). In addition, these four immunization groups produced significantly higher amounts of IL-21, an essential cytokine necessary for $B_{GC}$ cell help (AIM$^+$IL-21$^+$, $P = 0.020$, 0.0013, 0.0011, 0.0022, respectively; Fig. 2k–l). Interestingly by week 10, kinetics had shifted, with pSer:alum-3M-052 significantly producing higher amounts of IL-2 ($P = 0.041$), IFNγ ($P = 0.0044$), and granzyme B ($P = 0.045$) compared to pre-immunization (Fig. 2j, Supplementary Fig. 3k, m, o, q).

We also measured Env-specific circulating $T_{FH}$ (c$T_{FH}$, CXCR5$^+$ of CD4$^+$CD8a$^-$; Fig. 2m, Supplementary Fig. 3d). At week 2, all groups (alum, $P = 0.018$; pSer:alum, $P = 0.0005$; pSer:alum + SMNP, $P = 0.0019$; SMNP, $P = 0.0007$) produced significantly higher frequencies of Env-specific c$T_{FH}$ compared to pre-immune samples, except for pSer:alum-3M-052 ($P = 0.22$, Fig. 2n). However, by week 10, pSer:alum-3M-052 induced significantly higher Env-specific c$T_{FH}$ frequencies compared to baseline, indicating potential delayed kinetics of combined pSer:alum technology with 3M-052 (Fig. 2o). In sum, robust Env-specific CD4$^+$ T cells and c$T_{FH}$ were rapidly induced following priming immunization, with the most GC-$T_{FH}$ cells observed when adjuvanting with pSer:alum + SMNP.

## Combined pSer:alum approach with SMNP or 3M-052 enhances humoral responses

Considering the significant differences in $B_{GC}$ cell and GC-$T_{FH}$ cell responses between the adjuvant immunization strategies, we examined anti-Env serum IgG titers over time. pSer:alum + SMNP promoted faster antibody responses compared to all other groups: alum alone ($P < 0.0001$), pSer:alum ($P = 0.003$), SMNP alone ($P = 0.019$), and pSer:alum-3M-052 (week 2, $P = 0.0023$; Fig. 3a–b, Supplementary Fig. 4a). By week 6, pSer:alum augmented with either SMNP or 3M-052 generated more robust Env-binding IgG compared to alum alone ($P = 0.0073, 0.016$, respectively; Fig. 3c). Env IgG titers increased in all groups after the $1^{st}$ booster immunization. Notably, SMNP Group 4 exhibited the most stable antibody titers between weeks 12 to 24 compared to alum ($P = < 0.0001$, Fig. 3d, Supplementary Fig. 4b, c). Antibody boosts observed in all groups increased with the $3^{rd}$ and $4^{th}$ immunization (Fig. 3a). The data suggested that a durable high IgG response can be generated after one booster immunization when adjuvanting with SMNP, and two booster immunizations when adjuvanting with 3M-052.

Humoral immunity was also assessed by quantifying Env-specific bone marrow plasma cells (BM $B_{PC}$). At week 36, Env-specific BM $B_{PC}$ frequencies were significantly higher in animals immunized with SMNP, pSer:alum + SMNP, and pSer:alum-3M-052 compared to alum or pSer:alum (Fig. 3e). pSer:alum + SMNP induced a 13-times greater Env-binding BM $B_{PC}$ response compared to pSer:alum ($P = 0.0022$). Similarly, pSer:alum-3M-052 also elicited a 15-times greater response over alum ($P = 0.0022$) and a 12-times greater response over pSer:alum ($P = 0.0043$). SMNP Group 4 trended highest compared to all other immunization groups. Of note, SMNP induced a 53-times greater response than alum ($P = 0.0022$). The differences in Env-specific BM $B_{PC}$ responses persisted at week 49 (Fig. 3f, Supplementary Fig. 4d). In sum, robust and lasting antibody responses were differentially elicited between the adjuvant technologies used.

## pSer:alum + SMNP elicits high autologous neutralizing antibody titers

Development of nAb responses was assessed over time using TZM-bl autologous tier-2 BG505 HIV pseudovirus assays. By week 12, 5 out of 6 animals immunized with pSer:alum + SMNP developed nAbs, in contrast to 1 of 6 animals immunized with alum (1:365 geometric mean titer (GMT) vs. 1:12 GMT, $P = 0.015$; Fig. 4a). Post-$2^{nd}$ booster immunization, pSer:alum + SMNP immunized animals continued to trend higher than other groups (week 26), peaking two weeks post-$3^{rd}$ booster immunization at week 42 (Fig. 4b–c, Supplementary Fig. 5a). SMNP Group 4 also generated robust nAb responses at week 12, with GMTs approximately 20-times higher than immunization with alum ($P = 0.028$, 1:241 GMT; Fig. 4a). Equivalent results were found in neutralization assays performed independently (Supplementary Fig. 5b–e). Animals from any group with robust autologous tier-2 BG505 neutralization titers (ID$_{50}$ of 1:1,000 or higher) were tested on a global panel of heterologous tier-2 strains, revealing limited detectable neutralization (Supplementary Fig. 5f). Overall, pSer:alum + SMNP immunization conferred an advantage for induction of tier-2 HIV autologous nAbs.

## pSer:alum-mediated antigen delivery and SMNP adjuvant promote non-base-binding Env-specificities

Env antibody responses preferentially skew towards immunodominant non-neutralizing epitopes such as the base of the Env trimer[11,12]. We examined whether the pSer anchoring of Env trimer to alum masked the Env trimer base and shifted B cell recognition away from immunodominant non-neutralizing base responses. Sera were tested in cross-competition assays with 19 R, a high-affinity base-specific monoclonal Ab (mAb, Supplementary Fig. 6a)[12]. At week 6, there were no significant differences observed between any groups in their non-base-binding ability (Supplementary Fig. 6b). By week 10, all pSer groups − pSer:alum ($P = 0.0004$), pSer:alum + SMNP ($P = 0.0002$), and pSer:alum-3M-052 ($P = 0.0003$)−induced significantly more non-base-binding responses compared to alum alone (Fig. 5a, Supplementary

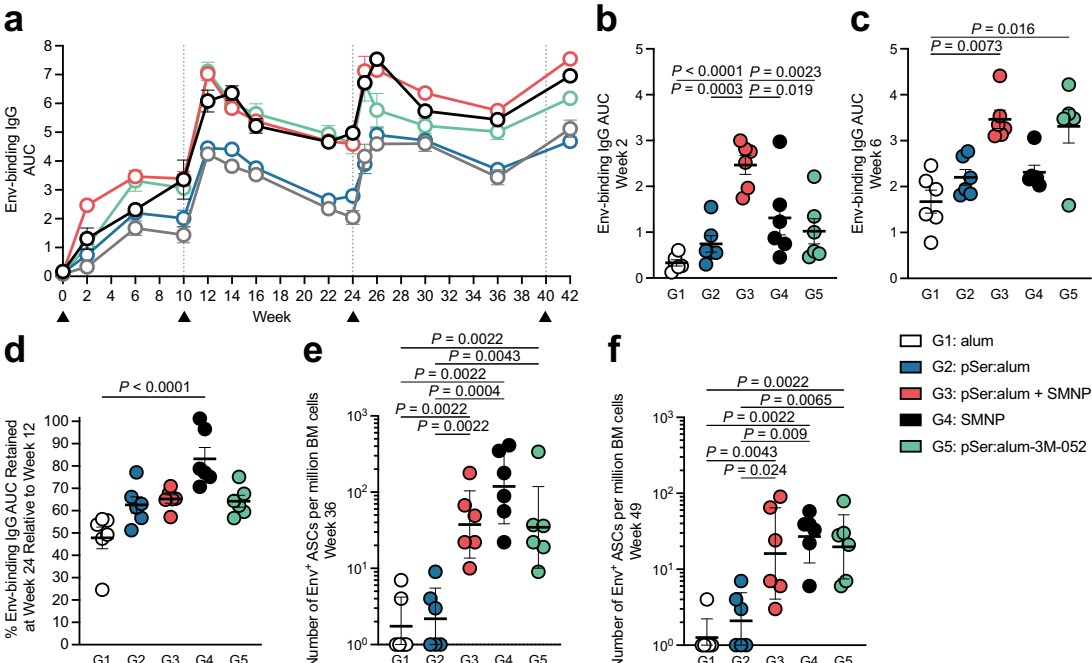

**Fig. 3 | Combined pSer:alum approach with either SMNP or 3M-052 induce enhanced antibody responses and promote bone marrow plasma cell responses. a** Full time course of the area under the curve (AUC) of total Env-binding serum IgG, as determined by ELISA. **b** AUC of Env-binding serum IgG at week 2. **c** AUC of Env-binding serum IgG at week 6. **d** Percentage of Env-binding AUC retained at week 24 relative to week 12. Percentage retained is calculated by taking the AUC at week 12 over the AUC at week 24. All other pairwise comparisons were not significant. **e** BG505 Env-specific bone marrow plasma cell (BM B$_{PC}$) responses from bone marrow aspirates as determined by ELISpot assays. Env-specific ELISpots were performed at week 36. **f** Env-specific ELISpots performed at week 49. Black triangles and gray dotted lines represent time of immunization.

Mean and SEM or geometric mean and geometric SD are plotted depending on the scale in all figures unless otherwise stated. For Fig. 3a–f, $n = 6$ per group. Statistical significance for **a**–**d** were measured using either Ordinary one-way ANOVA with Tukey's multiple comparisons test or Kruskal-Wallis test with Dunn's multiple comparisons test, depending on the distribution of the data (determined by normality or lognormality tests). Statistical significance for BM B$_{PC}$ ELISpots (**e**, **f**) was tested using either unpaired two-tailed Mann-Whitney tests or Kruskal-Wallis test with Dunn's multiple comparisons test, depending on the objectives of the study. *$p < 0.05$, **$p < 0.01$, ***$p < 0.001$, ****$p < 0.0001$. Source data are provided as a Source Data File.

Fig. 6a). pSer:alum + SMNP elicited the largest non-base response of any of the groups two-weeks post-1st booster immunization, significantly more than alum (Group 1, $P < 0.0001$) or pSer:alum (Group 2, $P = 0.0017$; Fig. 5b). This response is sustained to eight-weeks post-1st booster immunization (Supplementary Fig. 6c). Interestingly, non-base binding responses were also observed in SMNP Group 4 at frequencies above that of Group 1 (Fig. 5a–b). Overall, these results were consistent with pSer successfully modulating immunodominance of base epitopes.

To visualize the diversity of the serum antibody responses facilitated by different adjuvant technologies, we utilized electron microscopy polyclonal epitope mapping (EMPEM)[30,31]. EMPEM analysis of circulating antibodies suggested broader epitope targeting in animals receiving pSer:alum (Groups 2, 3, and 5) compared to alum, but the differences were not statistically significant (Fig. 5c, d, Supplementary Fig. 7a, b).

Lastly, to assess immunodominance at the cellular level, we quantified memory B cells (B$_{Mem}$, CD20$^+$IgD$^-$) in the periphery two weeks post-1st booster immunization to assess the degree to which the adjuvant technologies induced non-base-binding Env-specificities (Env$^+$Env-bKO$^+$), by utilizing an Env trimer with modifications to the base of the trimer to eliminate this immunodominant epitope (Env-bKO[21]; Fig. 5e, Supplementary Fig. 6d). Both pSer:alum + SMNP and SMNP elicited significantly higher frequencies of total Env-specific B$_{Mem}$ cells (Env$^+$) and on-target Env-specific B$_{Mem}$ cells (Env$^+$Env-bKO$^+$) compared to alum alone (Env$^+$, $P = 0.0022$, 0.0022; Env$^+$Env-bKO$^+$, $P = 0.0022$, 0.0022) and pSer:alum (Env$^+$, $P = 0.0043$, 0.0031; Env$^+$Env-bKO$^+$, $P = 0.0043$, $P = 0.022$; Fig. 5f–g, Supplementary Fig. 6e). In sum, pSer:alum stabilizes the Env trimer and, when used in conjunction with

potent adjuvants such as SMNP, reduced undesirable base-binding Env-specific IgG and B$_{Mem}$ cell responses.

## BCR sequencing of Env-binding B cells

Total Env-binding B cells from LN FNAs (Supplementary Fig. 2a, j) were also sorted for single cell B cell receptor (BCR) sequencing 6-weeks post-priming immunization at week 6, and 3-weeks after the booster immunizations at weeks 13 and 27 (Fig. 6a). To ensure sequences recovered were of antigen-experienced B cells, only class-switched immunoglobulin (Ig) sequences were considered for all BCR sequencing analyses. Somatic mutation analysis revealed successful recruitment of Env-binding B cells into germinal centers and continual affinity maturation, as evident by the increase of nucleotide (nt) mutations in the heavy chain (HC) from all immunization groups over time (median nt HC mutations at week 6, 13, and 27; Fig. 6b, Supplementary Fig. 8a–c). The class-switched sequences in each group contained a low percentage of unmutated sequences, ranging between 0.0 to 1.0% across all groups and timepoints (Fig. 6b). For all groups and timepoints, isotype IgG1 comprised at least 50% of all sequences, with the exception of pSer:alum post-boost, at week 13 and 27. At these timepoints, rhesus IgG4 contributions were substantial (26.9% IgG4 at week 13, 34.5% at week 27). Thus, pSer:alum appear to elicit a greater isotype diversity post-boost compared to all other immunization groups (Fig. 6c–e). Ranked clonal abundance curves post-priming immunization at week 6 indicated that pSer:alum + SMNP, SMNP Group 4, and pSer:alum-3M-052 elicited higher clonal diversity and greater number of clonal families compared to alum and pSer:alum (Fig. 6f). The data suggested successful GC recruitment and affinity maturation of

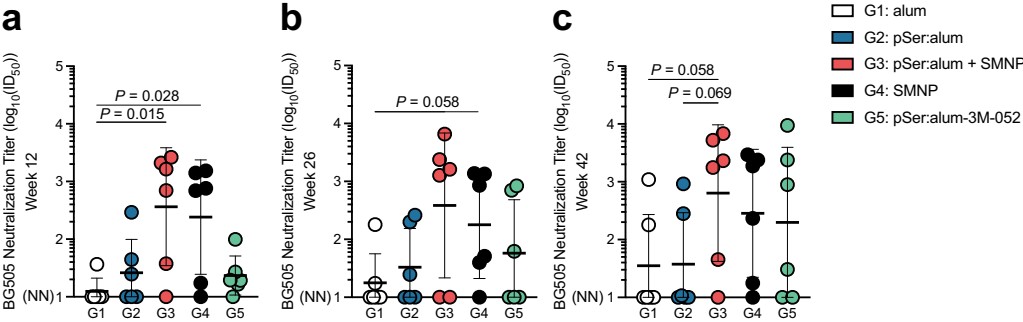

**Fig. 4 | pSer:alum+ SMNP elicits high autologous nAb titers.** Serum dilution at 50% inhibition of BG505 pseudovirus in a neutralization assay (ID$_{50}$). Neutralization assays were performed at: **a** week 12. **b** week 26. **c** week 42. NN = non-neutralizing. Neutralization data (Fig. 4a) for Group 1, alum, has been previously published[21]. Mean and SEM or geometric mean and geometric SD are plotted depending on the scale in all figures unless otherwise stated. For Fig. 4a–c, n = 6 per group, with two independent experiments averaged. Statistical significance was tested using either unpaired two-tailed Mann-Whitney tests or Kruskal-Wallis test with Dunn's multiple comparisons test, depending on the objectives of the study. *$p < 0.05$, **$p < 0.01$, ***$p < 0.001$, ****$p < 0.0001$. Source data are provided as a Source Data File.

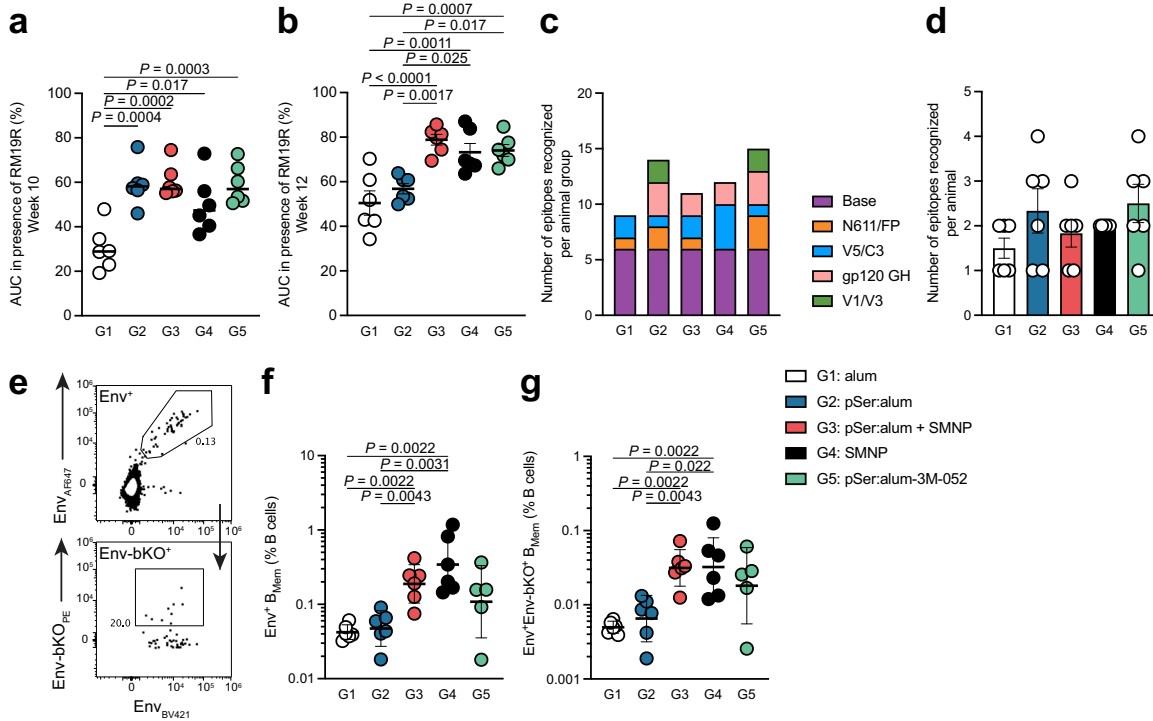

**Fig. 5 | pSer:alum-mediated antigen delivery and SMNP adjuvant promote non-base-binding Env-specificities. a** Antibody responses to the base of the trimer from serum at week 10, determined by cross-competition ELISA to 19R. Percentage of binding retained after addition of 19R is calculated by taking the AUC with 19R over the AUC without 19R. **b** Antibody responses to the base of the trimer, measured from serum at week 12. **c** Quantification of epitope sites on the Env trimer recognized using EMPEM, per animal group (out of 5 unique epitopes detected across all study groups), based on 3D maps presented in Supplementary Fig. 7a. **d** Total number of total epitope sites on the Env trimer recognized in EMPEM per animal, based on 3D maps presented in Supplementary Fig. 7a. **e** Flow cytometry gating of Env-specific (Env-AF647+Env-BV421+) and non-base-binding Env-specific (Env-AF647+Env-BV421+Env-bKO-PE+) memory B cells (B$_{Mem}$). **f** Frequency of Env-binding (Env+) cells as a percentage of total B cells. **g** Frequency of non-base-binding (Env+Env-bKO+) B$_{Mem}$ cells as a percentage of total B cells. EMPEM data (Fig. 5c-d) from Group 1, alum, have been previously published[21]. For Fig. 5a, b, f, g, n = 6 per group. Statistical significance for cross-competition ELISAs (a-b) was tested using Ordinary one-way ANOVA with Tukey's multiple comparisons test. Statistical significance for EMPEM and B$_{Mem}$ cell frequency (c-g) was tested using either unpaired two-tailed Mann-Whitney tests or Kruskal-Wallis test with Dunn's multiple comparisons test, depending on the objectives of the study. *$p < 0.05$, **$p < 0.01$, ***$p < 0.001$, ****$p < 0.0001$. Source data are provided as a Source Data File.

Env-binding B cells and that SMNP elicits substantial clonal diversity after priming compared to alum.

## Discussion
Understanding mechanisms that modulate B cell immunodominance is an imperative and ongoing aspect of rational vaccine design. Adjuvants may be a key factor impacting immunodominance in vaccines, by modulating the magnitude, breadth, and durability of adaptive immune responses to poorly immunogenic antigens and epitopes. Here, we demonstrate in an extensive head-to-head adjuvant comparison study that immunization with a combination of adjuvants, pSer:alum plus SMNP, can cooperate to substantially augment adaptive immune responses, particularly GCs.

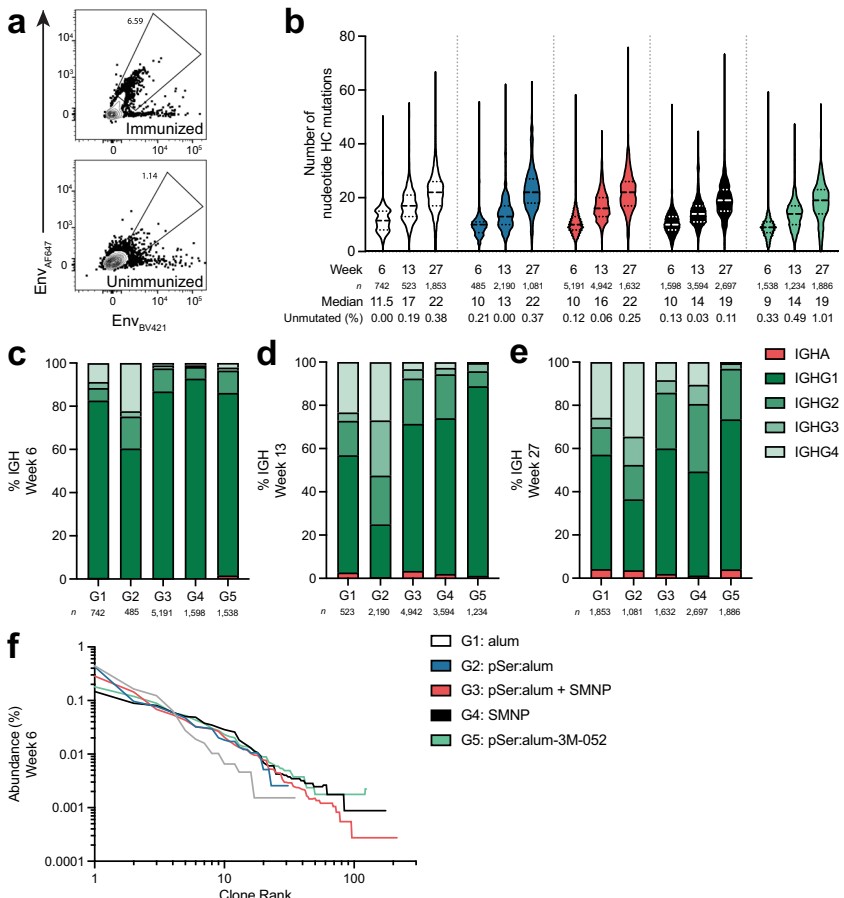

**Fig. 6 | BCR sequencing of Env-binding B cells. a** Flow cytometry gating of Env-binding B cells (Env-AF647⁺Env-BV421⁺) sorted for VDJ BCR sequencing. **b** The number of nucleotide heavy chain (HC) mutations over time of class-switched sequences. The percentage of unmutated sequences is calculated by taking the number of sequences with 0 nucleotide HC mutations over the number of total sequences. **c** The makeup of class-switched immunoglobulin (Ig) isotypes at week 6. **d** The makeup of class-switched Ig isotypes at week 13. **e** The makeup of class-switched Ig isotypes at week 27. **f** Clonal abundance curves of Env-binding B cells for each immunization group at week 6. For clonal abundance analysis (f), samples were excluded if fewer than 50 sequences were recovered. For Fig. 6a–f, sequences were collected from 12 samples (L and R LN, 6 animals per group). Source data are provided as a Source Data File.

GCs are sites of antibody affinity maturation and SHM, necessary for most neutralizing antibody responses. Here, direct sampling of the GC responses in the draining LNs over time revealed both robust $B_{GC}$ cell and GC-$T_{FH}$ cell responses from animals immunized with pSer:alum + SMNP post-priming immunization. A probable explanation for the enhanced $B_{GC}$ and Env-specific $B_{GC}$ cell responses at the priming immunization is that a combined immunization regimen not only allowed greater B cell access to antigen, but also promoted $B_{GC}$ cell differentiation by inducing greater numbers of GC-$T_{FH}$ cells and higher quality of GC-$T_{FH}$ cell support to $B_{GC}$ cells. Notably, Env-specific CD4+ T cells and Env-specific TFH cells were rapidly detected after priming immunization and appear to be functional, producing IL-2, IL-21, IFNγ, TNFα, and granzyme B.

The GC response advantages observed post-prime induced by pSer:alum + SMNP were not maintained with subsequent booster immunizations. Groups where GC and antibody responses were low in comparison to pSer:alum + SMNP at prime appear to increase and become comparable after multiple booster immunizations. Given the importance of durability in vaccine-induced immune responses, we postulate that having multiple booster immunizations – especially with potent adjuvants such as SMNP and 3M-052 – can help increase antibody and GC responses after a less optimal priming immunization. Nevertheless, the priming immunization is particularly important, as a series of studies have indicated that the priming immunization is the primary opportunity for recruiting rare precursor B cells—such as

autologous nAb-capable B cells and bnAb-class precursors—and booster immunizations usually do not compensate for insufficient recruitment of rare precursors at the initial priming immunization[12,15,25]. This idea is consistent with the enhanced induction of nAb responses for the pSer:alum + SMNP group compared to the other adjuvant conditions. Thus, the improved post-prime GC responses to pSer:alum + SMNP are notable. Somatic mutation analysis of Env-binding B cells revealed that the accrual of SHM in LNs was predominantly a function of time in GCs, with no significant modulation of SHM rate by adjuvants. The distribution of mutations and the low fraction of unmutated sequences at each timepoint, including after booster immunizations, indicated the vast majority of the B cell response was via continuing GCs or new GCs initiated by affinity-matured $B_{Mem}$ cells.

GC kinetics showed a differential induction of Env-specific $B_{GC}$ cells between adjuvant groups. We observed SMNP preferentially induces a larger Env-specific $B_{GC}$ cell response over conventional alum, possibly by initiating signals that sustains GCs longer thereby promoting $B_{GC}$ cell differentiation more effectively than other adjuvants. The current literature on $B_{GC}$ cell versus $B_{Mem}$ cell differentiation biases induced by different adjuvants is sparse, but understanding these mechanisms that underlie this critical decision-making process will be paramount in rational vaccine design.

One result of the GC reaction is affinity-matured antibodies. Early differences in Env-specific IgG titers observed between pSer:alum +

SMNP and all other groups likely result from low-affinity short-lived plasma cells exported from the GCs rapidly after immunization. SMNP can also enhance the durability of circulating antibodies, as IgG titers were sustained in SMNP-immunized animals. A look into the bone marrow at 9 and 11 months after the priming immunization revealed robust Env-specific BM $B_{PC}$ in animals that received pSer:alum with either SMNP or 3M-052 or SMNP itself, compared to alum and pSer: alum. The modest numbers of Env-specific BM $B_{PC}$ detected imply that immunization with alum may not induce BM $B_{PC}$ formation for poorly immunogenic, heavily glycosylated immunogens such as the MD39 trimer, and that the pSer-modification alone does not largely improve BM $B_{PC}$ responses. Rather, a second potent adjuvant, such as SMNP or 3M-052, is necessary to drive plasma cell responses. The detection of antigen-specific BM $B_{PC}$ responses at these late timepoints indicate some durability elicited from the adjuvant technologies. Efforts to circumvent immunodominance to promote nAb development continue to prove difficult. The robust Env-binding IgG titers observed from pSer:alum + SMNP and SMNP were not predictive of autologous nAb titers. Although SMNP-immunized animals produced higher autologous nAb titers compared to alum just after two immunizations, the effect was not observed at later time points. Subsequent booster immunizations did not prompt more RMs to make nAb responses. One hypothesis is that if the nAb-specific B cells are not sufficiently primed early, non-nAb B cells will continue to outcompete nAb-specific B cells later, irrespective of the number of booster immunizations[15].

An appeal of pSer-modification of antigen is the engagement of multiple adjuvant mechanisms of action to promote adaptive immune responses and to modulate immunodominance. Indeed, animals immunized with pSer-modified antigens shifted IgG responses away from the base of the Env trimer, potentially allowing for broader epitope targeting. Likewise, $B_{Mem}$ cells from animals that received pSer: alum and either SMNP or 3M-052 targeted potentially more favorable epitopes. Surprisingly, SMNP Group 4 appeared to overcome and compensate for some base-specific responses following boosting, possibly by enhancing antigen trafficking and secondary LN engagement to activate vast numbers of antigen-specific B cells targeting subdominant epitopes[9].

We have shown that immunization with multiple adjuvants can have an additive effect to promote adaptive immune responses. pSer:alum can mask immunodominant epitopes, while SMNP can improve B cell access to antigen and GC-$T_{FH}$ cell help to allow an added advantage over alum. 3M-052 encapsulated in biodegradable polymer particles at a much larger dose of 75 to 750 µg and alum-adsorbed 3M-052 at 75 µg have been previously used, either by itself or in combination with a second adjuvant TLR-4 agonist glucopyranosyl lipid adjuvant (GLA)[28,29]. We elected here to use a clinically relevant dose of 3M-052 (the 5 µg dose) and alum-adsorbed formulation used in an HIV vaccine human clinical trial (NCT04177355). In agreement with previous studies, our combined approach with multiple adjuvants allowed a significant advantage over alum, resulting in remarkable differences in the ability to elicit Env-specific plasma cell responses[28].

In HIV vaccine studies where animals received TLR-4 agonists formulated with alum or other synthetic agonists, greater humoral responses were induced over conventional alum and thus appear to be even more effective when used as a co-adjuvant[7,28]. In line with previous studies, SMNP is an ISCOMs-type adjuvant with TLR-4 agonist MPLA that robustly promoted adaptive immune responses compared to alum, when used either by itself or with pSer:alum. Comparisons between pSer:alum-3M-052 and pSer:alum + SMNP revealed comparable antibody and BM $B_{PC}$ responses. However, total $B_{GC}$ cells, Env-binding $B_{GC}$ cells, and early Env-specific T cells were all substantially higher in pSer:alum + SMNP immunized animals after the priming immunization. These different outcomes may be explained by different adjuvant mechanisms of actions and kinetics either intrinsic to the adjuvant or when combined with pSer:alum, or the lower 3M-052-alum dose (5 µg 3M-052/500 µg alum) compared to pSer:alum + SMNP (1000 µg alum/375 µg SMNP). Regardless, alum-adsorbed 3M-052 with pSer was still effective at a dose substantially lower than previously tested[28,29]. Recently, multiple SARS-CoV-2 studies using 3M-052-alum or 3M-052 in a squalene emulsion at dose levels of 5 to 10 µg reported robust adjuvant effects, eliciting strong neutralizing antibody titers and protection against SARS-CoV-2 in the respiratory tract[32–35]. For more complex, heavily glycosylated antigens like Env trimers, a combination of adjuvant approaches like pSer:alum with potent adjuvants with TLR agonists like SMNP and 3M-052 may be essential to mount neutralizing and protective responses.

Vaccine-induced protection from mucosal viral exposures is an important end goal in the development of an antibody-based HIV vaccine. Previously, it was found that nAb titers were a correlation of protection against SHIV$_{BG505}$ viral challenge in NHPs. Animals that were immunized with BG505 Env trimer and produced an autologous serum ID$_{50}$ nAb titer of 1:500 or better were afforded ~90% protection from a medium-dose homologous SHIV$_{BG505}$ infection[36]. For this study, 4 out of 6 animals immunized with pSer:alum + SMNP developed peak nAbs with an ID$_{50}$ of 1:500 or better, in contrast to alum or pSer:alum groups with 1 out of 6 animals in each group. Given the data, based on the nAb titers elicited, we would anticipate better protection against tier-2 SHIV$_{BG505}$ pseudovirus infection for animals immunized with BG505 Env trimer plus pSer:alum and SMNP compared to BG505 Env trimer plus alum. It is clear that the interplay of immunodominance factors is complex and the extent of their involvement in modulating adaptive immunity is still understudied. Further investigation into the impact of multiple adjuvants and mode of delivery on immunodominance in RMs would be a worthy undertaking. The most beneficial impact of the combined adjuvant approach occurred upon priming immunization. As priming is most important for the recruitment of rare B cells, pSer:alum with a potent adjuvant such as SMNP, may be a promising approach to initiate an environment that allows successful recruitment and competition by B cells that target subdominant neutralizing epitopes.

## Methods
### Immunogens
Three types of recombinant MD39 HIV Env gp140 trimers were produced: MD39 trimer with (a) C-terminal histag; (b) C-terminal histag followed by Cys residue for pSer conjugation; (c) C-terminal histag followed by avitag for biotinylation and use in cell sorting. All MD39 trimers were produced by transient transfection of HEK293F cells and co-transfection with Furin, purified by Ni affinity chromatography and size exclusion chromatography, characterized by SEC/MALS, endotoxin measurement (Charles River), and biotinylation level as appropriate, and then frozen, as previously described[24]. All MD39 trimer immunogens had <5.0 EU/mg endotoxin. pSer-conjugated MD39 was prepared as previously described[22]. In brief, a maleimide-functionalized PEG-(pSer)$_4$ peptide was generated by solid phase synthesis and purified by HPLC. This pSer tag was conjugated to MD39 trimers bearing a free Cys residue following the C-terminal purification histag via maleimide-thiol coupling and purified by centrifugal filtration. Successful functionalization was confirmed by a malachite green phosphate assay that detected 4.33 ± 0.75 phosphates per Env protomer on average. Alum binding of immunogens was measured using AlexaFluor647 NHS ester labeled immunogens. Antigenicity profiling of immunogens was completed by comparing antibody binding curves of pSer-conjugated MD39 on alum against those of unmodified MD39. To capture alum on Nunc Maxisorp ELISA plates, plates were first coated with pSer$_4$-conjugated cytochrome C at 2 µg/ml for 4 h at 25 °C. Alum was then added at 200 µg/ml and captured by pSer$_4$-cytochrome C overnight at 4 °C. To capture unmodified MD39, plates were coated with mouse VRC01 at 2 µg/ml for 4 h at 25 °C and blocked overnight at 4 °C with 2% BSA in PBS. Plates were washed with 0.05% Tween-20 in

PBS and incubated with 2 µg/ml MD39 in 2% BSA in PBS for 2 h at 25 °C. Neutralizing and non-neutralizing antibodies were added at 5 µg/ml with 1:4 serial dilutions for 2 h at 25 °C. Plates were washed and antibody binding was detected with a goat anti-human HRP conjugated secondary antibody with minimal cross-reactivity (Jackson ImmunoResearch) at 1:5000 dilution in PBS containing 2% BSA and then developed with 3,3′,5,5′-tetramethylbenzidine (Thermo Fisher), stopped with 2 N sulfuric acid and immediately read (450 nm with 540 nm reference) on a BioTek Synergy2 plate reader.

## Adjuvants and vaccine formulations
Alhydrogel was obtained from InvivoGen and used as received. SMNP adjuvant was prepared as previously described[9]. Alum-adsorbed 3M-052 was prepared by the Access to Advanced Health Institute (AAHI) as previously described[34] and was used as received. Immunizations were prepared by diluting immunogens in PBS and mixing with alum or alum with adsorbed 3M-052, as indicated. After a 20-minute incubation at 25 °C on a tube rotator to allow immunogens to adsorb to alum, SMNP was added to the indicated immunizations.

## Animals and immunizations
Indian rhesus macaques (RMs, *Macaca mulatta*) were housed at the Tulane National Primate Research Center and maintained in accordance with NIH guidelines. This study was approved by the Tulane University Institutional Animal Care and Use Committee (IACUC). Animals were grouped to match age, weight, and sex. All animals were between 3.5–5 years old at the time of the priming immunization, with each study group consisting of 3 females and 3 males ($n = 6$/group). A power analysis was previously performed to determine the optimal sample size to distinguish meaningful statistical differences in nAb titers between animal groups[25].

All immunizations were given subcutaneously (s.c.) in the left and right mid-thighs. Animals were immunized per side according to their respective study group, as follows: (i) Group 1 (MD39 + alum) with 50 µg MD39 and 500 µg alum (Alhydrogel adjuvant 2%, InvivoGen); (ii) Group 2 (pSer-MD39:alum) with 50 µg pSer-MD39 and 500 µg alum; (iii) Group 3 (pSer-MD39:alum + SMNP) with 50 µg pSer-MD39, 500 µg alum, and 187.5 µg SMNP; (iv) Group 4 (MD39 + SMNP) with 50 µg MD39 and 187.5 µg SMNP; and (v) Group 5 (pSer-MD39:alum + 3M-052) with 50 µg pSer-MD39 with 2.5 µg 3M-052 adsorbed onto 250 µg alum (IDRI-AL032). Doses of SMNP are reported in terms of the amount of saponin administered, as previously described[9]. Data from Group 1 have been previously published[21].

## Lymph node fine needle aspiration
Lymph node fine needle aspirates (LN FNAs) were performed by a veterinarian to sample the left and right draining inguinal LNs (iLNs), which were identified by palpation. The biopsy site was aseptically prepared and a 22-gauge 1.5-inch needle attached to a 3-mL syringe was passed into the LN 4–5 times. Samples were placed into RPMI containing 10% fetal bovine serum (FBS) and 1X penicillin/streptomycin (pen/strep). Ammonium-Chloride-Potassium (ACK) lysing buffer was used if the sample was contaminated with red blood cells. LN FNA cells were counted and divided between different assays; any extra cells were frozen down and stored in liquid nitrogen until further analysis.

## Bone marrow aspiration
Bone marrow samples were collected from the humerus or femur. The collection site was aseptically prepared and an 11-gauge Jamshidi needle was used to penetrate either the humerus or the femur. The stylet was removed from the needle, then a 6 ml syringe was attached to the Jamshidi needle to aspirate the bone marrow sample. Samples were suspended in RPMI containing 10% FBS and 1X pen/strep for ELISpot analysis.

## Flow cytometry
While LN FNA samples were stained fresh immediately after sample collection, frozen PBMC samples were thawed and recovered in RPMI media with 10% FBS, supplemented with 1X penicillin/streptomycin and 1X GlutaMAX (R10). Samples were stained with the appropriate antibody panel.

Fluorescent antigen probes were generated by mixing small incremental volumes of fluorophore-conjugated streptavidin with biotinylated MD39 (WT MD39) or MD39-base knockout (MD39-bKO) probes in 1x PBS at room temperature (RT) over 45 min. Cells were incubated with the WT MD39 probes for 30 min at 4 °C and then with surface antibodies for an additional 30 min at 4 °C. Where MD39-bKO probes were used, cells were first incubated with MD39-bKO probes for 20 min at 4 °C, then WT MD39 probes for 30 min at 4 °C, and finally with the surface antibodies for 30 min at 4 °C, similar to previously described protocols[12,21]. For samples being sorted, anti-human Total-Seq-C hashtag antibodies (BioLegend) were added to each individual sample at a concentration of 2 µg per 5 million cells along with the surface antibody master mix. All samples were either acquired on a FACSAria Fusion (BD Biosciences), a LSRFortessa (BD Biosciences), or a Cytek Aurora (Cytek Biosciences), depending on the experiment, or sorted on a FACSAria Fusion (BD Biosciences). The indexed V(D)J, Feature Barcode and Gene Expression libraries of sorted LN FNA were prepared following the protocol for Single Indexed 10X Genomics V(D)J 5′ v.1.1, with Feature Barcoding kit (10X Genomics). Custom primers were designed to target RM BCR constant regions. Primer set for PCR 1: forward, AATGATACGGCGACCACCGAGATCTACACTCTTTCCCTACACGACGCTC; reverse, AGGGCACAGCCACATCCT, TTGGTGTTGCTGGGCTT, TGACGTCCTTGGAAGCCA, TGTGGGACTTCCACTGGT, TGACTTCGCAGGCATAGA. Primer set for PCR 2: forward, AATGATACGGCGACCACCGAGATCT; reverse, TCACGTTGAGTGGCTCCT, AGCCCTGAGGACTGTAGGA, AACGGCCACTTCGTTTGT, ATCTGCCTTCCAGGCCA, ACCTTCCACTTTACGCT. Forward primers were used at a final concentration of 1 µM and reverse primers at 0.5 µM, each per 100 µl of PCR reaction, as previously described[21].

For bulk GC data inclusion in the LN FNA samples, a threshold of 250 total B cells in the sample was used. For Env-specific $B_{GC}$ cell data inclusion, a threshold of 75 total $B_{GC}$ cells was used. Any sample with fewer than 75 $B_{GC}$ cells, but with a B cell count greater than 500 cells was set to a baseline of 0.001% Env$^+$ $B_{GC}$ cells (as a percentage of total B cells). Otherwise, the limit of detection was calculated based on the median of (3/(number of B cells collected)) from the LN FNA samples at the pre-immunization timepoint. For GC-$T_{FH}$ cells, a threshold of 500 CD4$^+$ T cells was used. The $B_{GC}$ to GC-$T_{FH}$ cell ratio was calculated by taking the total number of $B_{GC}$ cells over the total number of GC-$T_{FH}$ cells, if the sample contained at least 500 B cells and 500 CD4$^+$ T cells.

For Env-bKO$^+$ (as a percentage of Env$^+$ $B_{Mem}$) calculations, a threshold of 10 Env$^+$ $B_{Mem}$ cells were used. PBMCs from NK03 from Group 5, pSer:alum-3M-052, was not collected and therefore not included in $B_{Mem}$ cell analysis.

The following reagents were used for staining (Supplementary Tables 1–2): Alexa Fluor 647 streptavidin (Invitrogen), BV421 streptavidin (BioLegend), PE streptavidin (Invitrogen), eBioscience Fixable Viability Dye eFluor 506 (Invitrogen, 1:500), LIVE/DEAD Fixable Aqua (Invitrogen, 1:1000), mouse anti-human CD3 BV786 (SP34-2, BD Biosciences, 1:67), mouse anti-human CD3 APC-Cy7 (SP34-2, BD Biosciences, 1:100), mouse anti-human CD4 BV650 (OKT4, BioLegend, 1:100), mouse anti-human CD8a APC-eFluor 780 (RPA-T8, Thermo Fisher Scientific, 1:200), mouse anti-human CD14 APC-Cy7 (M5E2, BioLegend, 1:100), mouse anti-human CD16 APC-eFluor 780 (eBioCB16, Thermo Fisher Scientific, 1:100), mouse anti-human CD16 APC-Cy7 (3G8, BioLegend, 1:100), mouse anti-human CD20 Alexa Fluor 488 (2H7, BioLegend, 1:50), mouse anti-human CD20 BUV395 (2H7, BD Biosciences, 1:100), mouse anti-human CD27 PE-Cy7 (O323,

BioLegend, 1:50), mouse anti-human CD38 PE (OKT10, NHP Reagents, 1:20), mouse anti-NHP CD45 BUV395 (D058-1283, BD Biosciences, 1:100), mouse anti-human CD71 PE-CF594 (L01.1, BD Biosciences, 1:20), mouse anti-human PD-1 BV605 (EH12.2H7, BioLegend,1:20), mouse anti-human CXCR5 PE-Cy7 (MU5UBEE, Thermo Fisher Scientific, 1:20), goat anti-human IgD Alexa Fluor 488 (polyclonal, Southern Biotech, 1:50), mouse anti-human IgG Alexa Fluor 700 (G18–145, BD Biosciences, 1:40), mouse anti-human IgG BUV737 (G18–145, BD Biosciences, 1:100), mouse anti-human IgM PerCP-Cy5.5 (G20–127, BD Biosciences, 1:40), mouse anti-human IgM BV605 (G20–127, BD Biosciences, 1:50) and TotalSeq-C anti-human Hashtag antibody 1–10 (LNH-94 and 2M2, Biolegend, 4 μL).

### Activation-induced marker (AIM) and intracellular cytokine staining (ICS) assay to detect antigen-specific CD4$^+$ T cells

Antigen-induced marker-based detection of antigen-specific T cells was performed similar to previously described protocols[37,38]. Cryo-preserved PBMCs were thawed and washed in R10 media. Cells were counted and then seeded at 1 million cells per well in a round-bottom 96-well plate. Before the addition of any stimulation condition, cells were blocked with 0.5 μg/mL anti-CD40 mAb (Miltenyi Biotec) and incubated with anti-CXCR5 and CCR7 for 15 min at 37 °C. Cells were then stimulated for 24 h with one of the following conditions: (1) 5 μg/mL MD39 Env peptide pool "Env"; (2) 1 ng/mL staphylococcal enter-otoxin B (SEB) used as a positive control "Pos"; or (3) DMSO as a negative, unstimulated control "Neg" plated in duplicate. MD39 Env peptide pools consist of overlapping 15-mer peptides that cover the entire protein sequence and were resuspended in DMSO. An equimolar amount of DMSO is present in both the peptide pool and the unsti-mulated, negative control. After 24 h of incubation, intracellular transport inhibitors – 0.25 μL/well of GolgiPlug (BD Biosciences) and 0.25 μL/well of GolgiStop (BD Biosciences) – were added to the sam-ples along with the AIM marker antibodies (CD25, CD40L, CD69, OX40, 4-1BB) and incubated for 4 h. After, the cells were washed and stained with the surface antibodies for 30 min at 4 °C. Briefly, cells were fixed with 4% formaldehyde and permeabilized with a saponin-based buffer and subsequently stained with the intracellular cytokine panel for 30 min at RT. Finally, the stained cells were washed and acquired on the Cytek Aurora (Cytek Biosciences).

For antigen-specific T cell data analysis, all test samples were calculated as background subtracted data, where the linear averages of the DMSO background signal from duplicate wells of the same sample were subtracted from the stimulated signal (signal–"Neg" DMSO background). A minimum threshold for DMSO background signals was set at 0.005% and the limit of quantitation (LOQ) was defined as the geometric mean of all "Neg" DMSO samples. For each test sample, the stimulation index (SI) was calculated as the ratio of the frequency of AIM$^+$ cells in the "Env" stimulated condition over the linear average "Neg" DMSO response in the same sample. Samples with an SI lower than 2 and/or with a background subtracted response lower than the LOQ were considered as non-responders. Non-responder samples were set at the baseline, which is the closest log$_{10}$ value lower than the LOQ. Sample NK04 from Group 3, pSer:alum + SMNP, was excluded from antigen-specific T cell analysis at week 2 and week 10 due to the viability of the cells.

The following reagents were used in the AIM and ICS assay (Supplementary Table 3): LIVE/DEAD Fixable Blue (Invitrogen), Golgi-Plug (BD Biosciences), GolgiStop (BD Biosciences), mouse anti-human CD40 (HB14, Miltenyi), mouse anti-human CXCR5 PE-Cy7 (MU5UBEE, Thermo Fisher Scientific), mouse anti-human CCR7 BV650 (G043H7, BioLegend), mouse anti-human CD69 PE-Cy5 (FN50, BioLegend), mouse anti-human CD137 (4-1BB) BV421 (4B4-1, BioLegend), mouse anti-human CD25 BV605 (BV96, BioLegend), mouse anti-human CD40L BB515 (24–31, BD Biosciences), mouse anti-human CD134 (OX40) PE (L106, BD Biosciences), mouse anti-human CD8 BUV496

(RPA-T8, BD Biosciences), mouse anti-human CD14 APC-Cy7 (M5E2, BioLegend), mouse anti-human CD16 APC-eFluor 780 (eBioCB16, Thermo Fisher Scientific), mouse anti-human CD20 APC-Cy7 (2H7, BioLegend), mouse anti-human CD3 BUV395 (SP34-2, BD Biosciences), mouse anti-human CD4 PerCP-Cy5.5 (OKT-4, BioLegend), mouse anti-human PD-1 BV785 (EH12.2H7, BioLegend), mouse anti-human CD45RA PE-CF594 (5H9, BD Biosciences), Armenian Hamster anti-ICOS BV480 (C398.4 A, BD Biosciences), mouse anti-human IFN-γ BUV737 (4 S.B3, BD Biosciences), rat anti-human IL-2 BV750 (MQ1-17H12, BD Bios-ciences), mouse anti-human TNF-α BV711 (Mab11, BioLegend), mouse anti-human Granzyme B Alexa Fluor 700 (GB11, BD Biosciences), mouse anti-human IL-21 Alexa Fluor 647 (3A3-N2.1, BD Biosciences), and human Fc block (Fc1, BD Biosciences).

### Neutralization assays

Pseudovirus neutralization assays at Scripps were performed as pre-viously described[25]. BG505 pseudovirus neutralization was tested using the BG505.W6M.ENV.C2 isolate with the T332N mutation to restore the N332 glycosylation site. Assays were done with duplicate wells per assay with independent repeats performed. Samples NK01, NJ85, NJ88, NK05 and NJ77 for week 42 autologous neutralization were run once (not repeated) due to sample availability. Heterologous neutralization breadth was tested on a panel of 13 cross-clade isolates, representative of larger virus panels isolated from diverse geography and clades[39]. They are listed as follows: MG505, CNE8, CNE55, TRO11, CH119, 246F3, 398F1, X1632, CEO217, X2278, 25710, CE1176, BJOX. Heterologous neutralization breadth assays at Scripps were performed with duplicate wells. The cut-off for neutralizing serum dilution was set at 1:10 depending on the starting serum dilution. Absolute ID$_{50}$ values were calculated using normalized relative luminescence units and a customized nonlinear regression model:

$$ID50 = Bottom + \frac{Top - Bottom}{1 + 10^{(LogAbsoluteIC50 - x)*Hill\ Slope + Log\left(\frac{Top - Bottom}{50 - Bottom} - 1\right)}}$$

with the bottom constraint set to 0 and top constraint set to the <100 model in Prism 8 (GraphPad).

Env-pseudotyped virus neutralization assays completed at Duke were measured as a function of reductions in luciferase (Luc) reporter gene expression after a single round of infection in TZM-bl cells[40,41]. TZM-bl cells (also called JC57BL-13) were obtained from the NIH AIDS Research and Reference Reagent Program, as contributed by John Kappes and Xiaoyun Wu. This is a HeLa cell clone that was engineered to express CD4 and CCR5[42] and to contain integrated reporter genes for firefly luciferase and E. coli beta-galactosidase under control of an HIV-1 LTR[43]. Briefly, a pre-titrated dose of virus was incubated with serial 3-fold dilutions of heat-inactivated (56 °C, 30 min) serum sam-ples in duplicate in a total volume of 150 μL for 1 h at 37 °C in 96-well flat-bottom culture plates. Freshly trypsin zed cells (10,000 cells in 100 μL of growth medium containing 75 μg/ml DEAE dextran) were added to each well. One set of control wells received cells + virus (virus control) and another set received cells only (background control). After 48 h of incubation, 100 μL of cells was transferred to a 96-well black solid plate (Costar) for measurements of luminescence using the Brit elite Luminescence Reporter Gene Assay System (PerkinElmer Life Sciences). ID$_{50}$/IC$_{50}$ neutralization titers/concentrations are the dilu-tion (serum/plasma samples) or concentration (mAbs) at which rela-tive luminescence units (RLU) were reduced by 50% or 80% compared to virus control wells after subtraction of background RLUs from cells controls. Assay stocks of molecularly cloned Env-pseudotyped viruses were prepared by transfection in 293 T/17 cells (American Type Culture Collection) and titrated in TZM-bl cells as described. This assay has been formally optimized and validated[44] and was performed in com-pliance with Good Clinical Laboratory Practices, including participa-tion in a formal proficiency testing program[45].

## ELISpot

HIV BG505-specific antibody secreting cells (ASCs) were detected by Enzyme-Linked ImmunoSpot (ELISPOT) assay. Total IgG and BG505 Env trimer-specific IgG-producing plasma spot-forming cells (SFC) in RMs were measured using freshly isolated PBMCs and bone marrow (BM) lymphocytes from BM aspirates collected at 36 and 49-weeks post-vaccination. Multi-screen HA-filtered ELISPOT plates (Millipore Sigma) were first coated with either total IgG (IgG + IgA + IgM, Exalpha Biologicals) or Galanthus Nivalis Lectin (GNL, Vector Laboratories) overnight at 4 °C for the detection of total IgG or BG505 Env-specific IgG responses, respectively. The next day, plates were washed once with PBS-0.05% Tween-20 (PBS-T) buffer and three times with PBS. After washing, all plates were blocked with complete RPMI media in a 6% $CO_2$ incubator at 37 °C for 2 h. GNL-coated plates were further incubated with HIV BG505 Env trimer, MD39 (20 µg/mL) for an additional 90 min at 37 °C. Plates were washed with PBS, cells were plated in 3-fold serial dilutions, and incubated overnight at 37 °C. After incubation, total IgG plates were washed with PBS followed by PBS-T. GNL-MD39 plates were washed with PBS only. All plates were then incubated overnight with anti-monkey biotin-conjugated goat IgG (Exalpha Biologicals) diluted in PBS-T or PBS for total IgG and MD39 Env-coated plates, respectively. After incubation, plates were washed and further incubated with horseradish peroxidase avidin D (Vector Laboratories) at RT for 3 h in the dark. The last wash was performed with PBS-T followed by PBS. Plates were then developed with 3-amino-9-ethylcarbazole substrate (Sigma) containing N,N-dimethylformamide and 3% hydrogen peroxide ($H_2O_2$) in 0.1 M Na-Acetate Buffer (pH 5.0) for 8 min. Once spots were developed, plates were washed with distilled $H_2O$, dried overnight, and protected from light before counting. Spots were documented using high-resolution automated ELISPOT reader systems from Zeiss (Zellnet Consulting Inc.) The mean number of spots per duplicate well was calculated. The total IgG and MD39 Env-specific IgG positive SFCs were presented after subtracting values obtained from PBS-only negative controls.

## ELISA

To determine serum MD39 IgG titers, Corning Costar high-binding 96-well plates (Corning) were coated with Streptavidin (Thermo Fisher) at 2 µg/ml for 4 h at 25 °C in PBS and blocked with 2% BSA in PBS over-night at 4 °C. Plates were washed three times with 140 mM NaCl, 3 mM KCl, 0.05% Tween-20 detergent, 10 mM phosphate buffer, pH 7.4 (Calbiochem), and biotinylated MD39 was added at 2 µg/ml in 2% BSA in PBS for 2 h. The plates were washed three times, and serum dilutions (1:50 dilution followed by 1:200 dilution with 1:4 serial dilutions) in 2% BSA in PBS were incubated in the plate for 2 h. Plates were washed three times, incubated with a Peroxidase AffiniPure Goat Anti-Human IgG, Fcγ fragment specific secondary antibody (Jackson ImmunoResearch) at 1:10000 dilution in 2% BSA in PBS, and incubated at 25 °C for 1 h. Plates were washed four times, then developed with 1-Step™ Ultra TMB-ELISA Substrate Solution (Thermo Fisher Scientific). Reaction was stopped with 2 N of sulfuric acid (Ricca Chemical Company), and immediately read (450 nm with 540 nm reference) on a FlexStation 3 Molecular Devices plate reader.

## EMPEM

The details of serum and sample preparation to obtain polyclonal fabs for electron microscopy were previously described[21]. Briefly, IgG was isolated using Protein A (Cytiva) from 1 mL NHP sera (drawn at week 12 post first immunization). Papain (Sigma Aldrich) was used to digest IgG to antigen-binding fragments (Fab). Trimer-fab complexes were prepared and incubated for an overnight by mixing 15 µg of BG505 MD39 SOSIP with 1 mg of fab mixture (containing Fc and residual papain). On the next day, the complexes were purified using a Superdex 200 Increase 10/300 GL gel filtration column (Cytiva). Purified complexes

were concentrated and diluted to a final concentration of 0.03 mg/mL, which were adsorbed on glow-discharged carbon coated copper mesh grids and stained with 2% (w/v) uranyl formate. Electron microscopy images were collected on an FEI Tecnai Spirit T12 equipped with an FEI Eagle 4k x 4k CCD camera (120 keV, 2.06 Å/pixel) and processed using Relion 3.0[46] following standard 2D and 3D classification procedures. Leginon was used to automate EM data collection. UCSF Chimera v1.13[47] was used to generate the composite maps, and the representative maps with identified epitopes have been deposited to the Electron Microscopy Data Bank under accession codes listed in Supplementary Fig. 7b.

## BCR sequencing, processing, and analysis

Cellranger v6.1.2 was used for generating FASTQ files, performing full-length VDJ read assembly, and processing feature barcode data. The VDJ read assembly was performed in de novo mode in CellRanger, and later aligned to custom rhesus germline VDJ reference using the Change-O package v1.3.0 within the Immcantation Portal v4.4.0[12,48–50]. The Change-O pipeline parsed the 10X V(D)J sequence output from CellRanger into an AIRR community standardized format, to allow for more downstream analysis using packages from the Immcantation Portal. Sequences were de-multiplexed by hashtags using the MULTI-seqDemux command in Seurat v4[51]. For all analyses, only class-switched (IgG isotype subclasses and IgA) paired heavy and light chain sequences (with 95% identity in constant region calls) were considered.

For BCR analysis, we performed somatic hypermutation analysis, assessed isotype composition, and plotted ranked clonal abundance curves. Somatic hypermutation analysis was performed using the observedMutations command within the SHazaM package v1.1.2[50]. The total number of mutations (within V- and J-genes) for each heavy chain (HC) was determined by counting the number of nucleotide changes between the observed sequence and the predicted germline sequence. The inferred germline V and J sequences from the RM reference were added with CreateGermline.py within the Change-O package. The germline D gene sequences and N nucleotide additions were masked from analysis since these cannot be accurately predicted. For analyses performed on an individual animal basis, a minimum of 10 sequences were required. Animal samples that did not reach this threshold were excluded from analysis. The isotype was determined using VSEARCH v2.21.1[52], which matches the query VDJ sequences to a list of constant region Ig sequences. The sequences were then filtered for at least 95% identity in the constant region calls. Ranked clonal abundance curves were created by counting each unique clonal family per animal, using the countClones command within Alakazam v1.2.1[50], then graphing the relative abundance. For clonal abundance analysis, a minimum of 50 sequences recovered were used. Samples were excluded if this threshold was not reached.

## Graphs and statistical analysis

All statistics were calculated in Prism 9 or R unless stated otherwise. The statistical tests used were indicated in the respective figure legends and were used depending on the objectives of the study and the hypotheses at the start of the study. In brief, for $B_{GC}$ cell analysis, GC-$T_{FH}$ cell analysis, BM $B_{PC}$ analysis, BG505 Neutralization titers ($ID_{50}$), EMPEM, and $B_{Mem}$ cell analysis, either an unpaired two-tailed Mann-Whitney test (G1 vs. all groups, G2 vs. G3, G5) or a Kruskal-Wallis test with Dunn's multiple comparisons test (G2 vs. G4, G3 vs. G4 vs. G5) corrections were performed. For Env-specific T cell AIM assays, a Kruskal-Wallis test with uncorrected Dunn's multiple comparisons test was performed for all groups. For the Env-binding IgG ELISAs, either an Ordinary one-way ANOVA with Tukey's multiple comparisons test or Kruskal-Wallis test with Dunn's multiple comparisons test, depending on the distribution of the data (determined by Normality and Log-normality tests). For base-binding cross-competition ELISAs, an Ordinary one-way ANOVA with Tukey's multiple comparisons test was

performed. For pSer-MD39 characterization data, a two-way ANOVA with either Sidak's multiple comparisons test or Tukey's multiple comparisons test was performed. All graphs were generated in Prism 9. Geometric mean and geometric standard deviation (s.d.) were shown for data plotted on a $\log_{10}$ and $\log_2$ scale, while mean and standard error of the mean (SEM) or mean and s.d. were shown for data plotted on a linear scale. P values are defined throughout as follows: not significant, $P > 0.05$; *$P \leq 0.05$; **$P < 0.01$; ***$P < 0.001$; ****$P \leq 0.0001$.

## Reporting summary

Further information on research design is available in the Nature Portfolio Reporting Summary linked to this article.

## Data availability

The BCR sequencing data generated in this study are available in the Sequence Read Archive (SRA) under accession code PRJNA1016452. The 3D EM reconstructions are available from the Electron Microscopy Bank under the following EMDB codes: EMD-40242, EMD-40243, EMD-40244, EMD-40252, EMD-40254, EMD-40255, EMD-40256, and EMD-40257. Sequencing data and electron microscopy particle stacks are also available upon request. Source data are provided with this paper.

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

## Acknowledgements

We would like to thank the Sequencing Core Facility at the La Jolla Institute for Immunology. We thank J. Wood from Emory National Primate Research Center for technical guidance and feedback on the LN FNA technique. This work was funded by the National Institute of Allergy and Infectious Diseases of the NIH (NIAID-NIH) under award CHAVD UM1AI144462. The work at Tulane National Primate Research Center was supported under awards P51OD011104; the NIAID-NIH, Simian Vaccine Evaluation Unit Program (HHSN272201300004I - Task Order No. 75N93019F00131) coordinated by N. Miller; and the Cellular Immunology Laboratory Contract (HHSN27221700022C) coordinated by Q. Dang. Neutralization assays completed at Duke were supported by the NIAID-NIH Contract HSN272201800004C. D.J. Irvine is an investigator of the Howard Hughes Medical Institute and was supported by NIH award P01AI048240 (D.J.I).

## Author contributions

I.P. designed experiments, performed flow cytometry for all PBMC samples, and analyzed all data. K.A.R. performed pSer-MD39 conjugation and QC. E.M.Z. and P.G.L. performed AIM assays. L.M., M.M., and K.K.M. performed ELISA assays. B.P. performed bone marrow ELISpots and NGS library preparation. W.-H.L., J.L.T. and G.O., and A.B.W. performed EMPEM analysis. A.K., C.A., and M.F. processed LN FNA samples and ran flow cytometry. B.F.G. and J.P.D. performed animal immunizations and LN FNA collections. F.S., P.P.A. and R.S.V. coordinated and managed the animal study at Tulane. S.E., M.K., E.G., B.G., and W.R.S. provided immunogens and flow cytometry baits. R.N., M.B., L.H. and D.R.B. performed neutralization assays. H.G., X.S. and D.C.M performed independent confirmation neutralization assays. D.G.C. and G.S. provided technical guidance and samples used in QC in preparation for the study. I.P. and S.C. wrote the paper. D.J.I. and S.C. conceived and supervised the study. All authors provided comments and edits to manuscript.

## Competing interests

W.R.S is named as an inventor on a patent for the MD39 immunogen (US No. 11,203,617). D.J.I. and W.R.S. are named as inventors on a patent for pSer technology (US No. 11,224,648 B2). D.J.I and S.C. are inventors on a patent for the SMNP adjuvant (US No. 11,547,672 B2). K.A.R. and D.J.I. are inventors on patent applications for the synergistic combination of alum and SMNP adjuvants (PCT/US2022/074302 and US No. 17/816,045). The remaining authors declare no competing interests.

## Additional information

Ivy Phung [1,2,3], Kristen A. Rodrigues [4,5], Ester Marina-Zárate [1,2], Laura Maiorino[4,5], Bapi Pahar [6], Wen-Hsin Lee [7], Mariane Melo[2,4,5], Amitinder Kaur[6], Carolina Allers[6], Marissa Fahlberg[6], Brooke F. Grasperge[6], Jason P. Dufour[6], Faith Schiro[6], Pyone P. Aye [6], Paul G. Lopez[1], Jonathan L. Torres [7], Gabriel Ozorowski [2,7,8], Saman Eskandarzadeh[2,8,9], Michael Kubitz[2,8,9], Erik Georgeson [2,8,9], Bettina Groschel[2,8,9], Rebecca Nedellec[9], Michael Bick[9], Katarzyna Kaczmarek Michaels[4,5], Hongmei Gao [10], Xiaoying Shen [10], Diane G. Carnathan[11], Guido Silvestri [11], David C. Montefiori [10], Andrew B. Ward [2,7], Lars Hangartner [2,9], Ronald S. Veazey [6], Dennis R. Burton [2,5,8,9], William R. Schief [2,5,8,9], Darrell J. Irvine [2,4,5,12,13,14] ✉ & Shane Crotty [1,2,3] ✉

[1]Center for Infectious Disease and Vaccine Research, La Jolla Institute for Immunology (LJI), La Jolla, CA 92037, USA. [2]Consortium for HIV/AIDS Vaccine Development (CHAVD), The Scripps Research Institute, La Jolla, CA 92037, USA. [3]Department of Medicine, Division of Infectious Diseases and Global Public Health, University of California, San Diego (UCSD), La Jolla, CA 92037, USA. [4]Koch Institute for Integrative Cancer Research, Massachusetts Institute of Technology, Cambridge, MA 02139, USA. [5]Ragon Institute of Massachusetts General Hospital, Massachusetts Institute of Technology and Harvard University, Cambridge, MA 02139, USA. [6]Tulane National Primate Research Center, Tulane School of Medicine, Covington, LA 70433, USA. [7]Department of Integrative Structural and Computational Biology, The Scripps Research Institute, La Jolla, CA 92037, USA. [8]IAVI Neutralizing Antibody Center, The Scripps Research Institute, La Jolla, CA 92037, USA. [9]Department of Immunology and Microbiology, The Scripps Research Institute, La Jolla, CA 92037, USA. [10]Department of Surgery, Laboratory for AIDS Vaccine Research & Development, Duke University Medical Center, Duke University, Durham, NC 27710, USA. [11]Emory National Primate Research Center and Emory Vaccine Center, Emory University School of Medicine, Atlanta, GA 30322, USA. [12]Department of Biological Engineering, Massachusetts Institute of Technology, Cambridge, MA 02139, USA. [13]Department of Materials Science and Engineering, Massachusetts Institute of Technology, Cambridge, MA 02139, USA. [14]Howard Hughes Medical Institute, Chevy Chase, MD 20815, USA. ✉e-mail: djirvine@mit.edu; shane@lji.org

