## [Peer Review File · Nature Communications]

A combined adjuvant approach primes robust germinal center responses and humoral immunity in non-human primatesReviewers' Comments:

Reviewer #1:

Remarks to the Author:

The manuscript by Phung et al reports on the the benefit of a combined adjuvant approach to prime a robust germinal center response and humoral response in non-human primates. They evaluated various parameters associated with B cell responses and demonstrated that priming with pSer:alum+SMNP induced higher and envelope specific GC B cell and GC-TFH responses, as well as non-base binding neutralization antibodies. This type of research is important in the context of vaccine formulations that could induce and potentially boost desired immune responses for HIV-1 control.

The experiments were reasonably conducted, the conclusions drawn from this study are reasonable and are supported by the data, and the manuscript is well-written.

I recommend this manuscript be published

Comment

(i) The data shows that the "superior" germinal center B cell responses induced by pSer:alum+SMNP over the other immunization regimens was not maintained especially post-second boost. Would the responses induced post-prime (Fig. 1c) have persisted in the absence of boosting, taking into account the potency of ISCOMs?

The authors should discuss this drop in durability taking into account the importance of durable responses and immune outcomes.

(ii) Line 154. close the bracket after (Figure 1a

Reviewer #2:

Remarks to the Author:

Ivy Phung et al conducted an extensive head-to-head adjuvant comparison study in RM and asked whether specific combinations with the stabilized recombinant native-like HIV Env trimer MD39 were impacting Tfh/GC B cell reactions and more importantly, the B cell immunodominance. Alum is the most prevalent adjuvant used in licensed human vaccines but investigating other adjuvant formulation, together with antigen delivery is key for the development for HIV vaccine rational design.

This study is in the continuity of previous published studies of the authors who investigated immunization approaches using the MD39 immunogen, the pSer technology and the SMNP adjuvant (ISCOM incorporating TLR4 agonist MPLA). It was still unclear whether pSer modification may have significant impact on the kinetics of adaptive immune response and whether additional adjuvants may have synergistic effects. To address these questions, authors immunized five groups of RM (n=6/grp) administering either the gold-standard Alum (G1), pSer-MD39 bound to Alum (G2), pSer-Alum+SMNP (G3), soluble MD39 mixed with SMNP (G4), or pSer:Alum preabsorbed with 3M-052 TLR7/8 agonist (G5). Doses have been either established to be at lower concentrations than in previous NHP studies, or to reflect formulations in ongoing human vaccine clinical trial (NCT04177355). Fine-needle aspirations has been performed regularly to analyze the frequency of germinal center B cells by FACS. Post-prime, pSer:alum and SMNP work synergically to promote GC, which is also observed after the 1st booster but at a lesser extent, while no difference could be shown after the 2nd booster. Strikingly, authors observed a shift of total B cells towards Env-binding B GC with pSer:alum after the prime. Initial B GC cell differentiation was promoted together with a robust GC-Tfh cell help, which is known to be beneficial in the quality of GC-TFH cell help. Functionality of Tfh cells, with notably the production of IL-21 was also demonstrated by AIM-ICS assay on PBMCs, although no significant differences could be observed between groups. Together with the GC/Tfh reaction, the production of binding Env IgG appeared faster and durable when adjuvanting with SMNP. Strikingly, authors

quantified bone marrow plasma cells and demonstrated their persistence in groups immunized with SMNP and 3M-052, indicating a robust and long-lasting antibody response. Last, authors investigated NeutAb titers. Neutralizing assays, performed by 2 independent teams, confirmed an advantage of pSer:alum + SMNP for inducing tier-2 HIV autologous NeutAb, but unfortunately not for heterologous Tier-2 strains. Nevertheless, authors established competitive binding assays and staining of Env memory B cells using modified trimer, and obtained thus evidence that pSer:alum + SMNP reduces undesirable base-binding Env-specific IgG and B memory cell responses.

The work is a comprehensive study, very-well writing, with rationale of each experiment well given. Authors used advanced and appropriate techniques, and data have been carefully analyzed. Figures have been selected to highlight the quality of their data and all appropriate controls that are requested are presented in the supplementary figures. Methods are presented in detail. Data support conclusions of the authors, and any weakness of the study (EMPEM, NeutAb) is discussed honestly. Any complementary experiments would be out-of-the-scope of this specific study. Authors do not overestimate their conclusions.

Globally, results underscore the importance of potent adjuvant such as SMNP to prime GC/Tfh responses. We are aware that repeating trimeric Env might be necessary to induce potent NeutAb and therefore, the advantage of pSer:Alum + SMNP could be questionable. Authors discuss their critical findings, especially the fact that pSer:Alum + SMNP might be an advantage during the priming for limiting B cell immunodominance. This work will be of importance for future preclinical assay (eg testing mode of delivery of antigen). Vaccine tools and approaches are also relevant for the development of clinical trials.

Reviewer #3:

Remarks to the Author:

In this study conducted by Ivy Phung et al., the immunogenicity of five different HIV vaccine immunization types was examined, with variations in immunogen, adjuvants, and potential in-vivo antigen delivery kinetics. The researchers modified an HIV Env trimer by incorporating phosphoserine peptide linkers to improve its binding to aluminum hydroxide (pSer:alum). The study design involved a complex arrangement of 5 arms with 6 rhesus macaques per group, examining combinations of the original or modified trimers with three different adjuvants (alum, SMNP, 3M-052).

According to the work, the combination of pSer:alum with the adjuvant SMNP resulted in enhanced germinal center (GC) responses and GC-TFH cell help. Notably, the adjuvants 3M-052 and SMNP, in combination with pSer:alum, led to the presence of long-lasting Env-specific bone marrow antibody-secreting cells. Furthermore, modifying the trimer with pSer reduced the targeting of non-neutralizing epitopes.

While this manuscript is well-written and contributes to the exploration of different adjuvants in HIV vaccines, there are some important considerations to address. Briefly, the manuscript lacks experimental evidence to support the superiority of one approach over another and lacks of results that explain the mechanisms underlying the observed differences in immune responses. Specific comments are below.

1. The description of antigen delivery kinetics could be expanded, as it is currently limited to few words and a supplemental figure.
2. The statistical justification for having only six rhesus macaques per group is missing, raising questions about the ability to detect the reported immunological differences with this sample size. In fact, both Figure 1 and Figure 2 relied on values obtained from two lymph nodes collected from the same animals to achieve statistical significance. This approach of dividing the vaccine dose and administering it bilaterally is not translatable to humans, and the use of values from the same animal without proper identification might be improper. Are there any differences between values obtained

from two lymphnodes in the same animal? If yes, why? Why did researchers not plot the average of the results obtained from the two lymphnodes if the study was properly powered? The utilization of values obtained from two lymph nodes collected from the same animals to achieve statistical significance is questionable and may not accurately reflect biological duplicates. It would be beneficial to provide clarification on any potential differences between the values obtained from the two lymph nodes and consider plotting the average results or to show matching results.

3. The claim of superiority of one approach over another lacks support from experiments that could explain the mechanisms behind the observed "larger," "faster," and "more durable" immune responses. The designation of a faster antibody response based on a single measurement at week 2 seems arbitrary. Moreover, at week 4, when the antibody peak is reached and an objective threshold can be identified, no differences were observed between the pSer:alum + SMNP group and others, raising questions about the necessity of trimer modification and alum to achieve similar outcomes in clinical studies. The lack of significant differences in antibody quantity at week 10, despite differences in B- and T-cells at the same time point, further complicates interpretation.

4. It is important to note that the efficacy of these vaccines has not been demonstrated through mucosal viral challenges, which is a significant lack for readers of this journal. Indeed higher immunogenicity is not often translated in higher vaccine efficacy or vaccine efficacy at all in the context of strategies to prevent HIV.

5. Even accepting the lack of mucosal viral exposures and considering this as an immunogenic under-powered study it is a missed opportunity the lack of data of mucosal sites, where viral transmission happens.

Overall, while the manuscript is well-written, the study is highly descriptive with flaws in its design. We encourage further work to solidify these preliminary findings.

Reviewer #1 (Remarks to the Author):

The manuscript by Phung et al reports on the the benefit of a combined adjuvant approach to prime a robust germinal center response and humoral response in non-human primates. They evaluated various parameters associated with B cell responses and demonstrated that priming with pSer:alum+SMNP induced higher and envelope specific GC B cell and GC-TFH responses, as well as non-base binding neutralization antibodies. This type of research is important in the context of vaccine formulations that could induce and potentially boost desired immune responses for HIV-1 control.

The experiments were reasonably conducted, the conclusions drawn from this study are reasonable and are supported by the data, and the manuscript is well-written.

I recommend this manuscript be published

A: Thank you for your positive assessment. We have also added new data, where we performed BCR sequencing on Env-binding B cells sorted from the lymph node fine needle aspirates. We assessed somatic mutation rates and isotype compositions of class-switched sequences over time. We also performed ranked clonal abundance curves post-priming immunization at week 6. With this new data, we hope to fully round out this work and to be a comprehensive resource for future studies.

Comment

(i) The data shows that the "superior" germinal center B cell responses induced by pSer:alum+SMNP over the other immunization regimens was not maintained especially post-second boost. Would the responses induced post-prime (Fig. 1c) have persisted in the absence of boosting, taking into account the potency of ISCOMs?

The authors should discuss this drop in durability taking into account the importance of durable responses and immune outcomes.

A: Thank you for your comment and suggestion. We have recently shown that employing a slow delivery immunization strategy (e.g., escalating-dose immunization) with a potent ISCOMs adjuvant (SMNP) and allowing the germinal centers to continue undisturbed for over 6 months induced durable GCs and enhanced antibody quality (Lee, Sutton, et. al., *Nature* 2022). With that in mind, it is certainly possible that in the absence of boosting, animals primed with pSer:alum (slow delivery) + SMNP (potent ISCOMs adjuvant) could also have long-lasting GCs induced with enhanced neutralizing antibody responses. However, a more formal testing of this hypothesis is needed, as the combination of different immunogen delivery platforms with adjuvants may alter their mechanisms of action.

The reviewer is correct in that the GC response advantage observed post-prime induced by pSer:alum + SMNP were not maintained with subsequent booster immunizations. It is possible that having multiple booster immunizations can help compensate for a less optimal priming immunization. This was observed in SMNP Group 4 and pSer:alum-3M-052, where GC and antibody responses were quite low but increased with subsequent boosts. Bone marrow ELISpots show antigen-specific plasma cell responses at 9 and 11 months after multiple immunizations, indicating some durable responses elicited. This has been reflected in the revised Discussion section:

"GCs are sites of antibody affinity maturation and SHM, necessary for most neutralizing antibody responses. Here, direct sampling of the GC responses in the draining LNs over time revealed both robust B_{GC} cell and GC-T_{FH} cell responses from animals immunized with pSer:alum + SMNP post-priming immunization. A probable explanation for the enhanced B_{GC} and Env-specific B_{GC} cell responses at the priming immunization is that a combined immunization regimen not only allowed greater B cell access to antigen, but also promoted

B_{GC} cell differentiation by inducing greater numbers of GC-T_{FH} cells and higher quality of GC-T_{FH} cell support to B_{GC} cells. GC response advantages observed post-prime induced by pSer:alum + SMNP were not maintained with subsequent booster immunizations. Groups where GC and antibody responses were low in comparison to pSer:alum + SMNP at prime appear to increase and become comparable after multiple booster immunizations. Given the importance of durability in vaccine-induced immune responses, we postulate that having multiple booster immunizations - especially with potent adjuvants such as SMNP and 3M-052 - can help compensate for a less optimal priming immunization."

However, the data indicate that booster immunizations cannot fully compensate for shortcomings during priming, and thus the improved priming of GCs for pSer are notable. This is a topic for clarification also raised by Reviewer 2. We added further consideration of this topic in the Discussion: Nevertheless, the priming immunization is particularly important, as a series of studies have indicated that the priming immunization is the primary opportunity for recruiting rare precursor B cells—such as autologous nAb-capable B cells and bnAb-class precursors—and booster immunizations usually do not compensate for insufficient recruitment of rare precursors at the initial priming immunization^{12,15,24}. This idea is consistent with the enhanced induction of nAb responses for the pSer:alum + SMNP group compared to the other adjuvant conditions. Thus, the improved post-prime GC responses to pSer:alum + SMNP are notable.

We have commented on bone marrow plasma cell responses in the Discussion section and has been expanded further:

"A look into the bone marrow at 9 and 11 months after the priming immunization revealed robust Env-specific BM B_{PC} in animals that received pSer:alum with either SMNP or 3M-052 or SMNP itself, compared to alum and pSer:alum. The modest numbers of Env-specific BM B_{PC} detected imply that immunization with alum may not induce BM B_{PC} formation for poorly immunogenic, heavily glycosylated immunogens such as the MD39 trimer, and that the pSer-modification alone does not largely improve BM B_{PC} responses. Rather, a second potent adjuvant, such as SMNP or 3M-052, is necessary to drive plasma cell responses. The detection of antigen-specific BM B_{PC} responses at these late timepoints indicates some durability elicited from the adjuvant technologies."

(ii) Line 154. close the bracket after (Figure 1a

A: Thank you. This has been corrected.

Reviewer #2 (Remarks to the Author):

Ivy Phung et al conducted an extensive head-to-head adjuvant comparison study in RM and asked whether specific combinations with the stabilized recombinant native-like HIV Env trimer MD39 were impacting Tfh/GC B cell reactions and more importantly, the B cell immunodominance. Alum is the most prevalent adjuvant used in licensed human vaccines but investigating other adjuvant formulation, together with antigen delivery is key for the development for HIV vaccine rational design.

This study is in the continuity of previous published studies of the authors who investigated immunization approaches using the MD39 immunogen, the pSer technology and the SMNP adjuvant (ISCOM incorporating TLR4 agonist MPLA). It was still unclear whether pSer modification may have significant impact on the kinetics of adaptive immune response and whether additional adjuvants may have synergistic effects. To address these questions, authors immunized five groups of RM (n=6/grp) administering either the gold-standard Alum (G1), pSer-MD39 bound to Alum (G2), pSer-Alum+SMNP (G3), soluble MD39 mixed with SMNP (G4), or pSer:Alum preabsorbed with 3M-052 TLR7/8 agonist (G5). Doses have been either established to be at lower concentrations than in previous NHP studies, or to reflect formulations in ongoing human vaccine clinical trial (NCT04177355). Fine-needle aspirations has been performed regularly to analyze the frequency of germinal center B cells by FACS. Post-prime, pSer:alum and SMNP work synergically to promote GC, which is also observed after the 1st booster but at a lesser extent, while no difference could be shown after the 2nd booster. Strikingly, authors observed a shift of total B cells towards Env-binding B GC with pSer:alum after the prime. Initial B GC cell differentiation was promoted together with a robust GC-Tfh cell help, which is known to be beneficial in the quality of GC-TFH cell help. Functionality of Tfh cells, with notably the production of IL-21 was also demonstrated by AIM-ICS assay on PBMCs, although no significant differences could be observed between groups. Together with the GC/Tfh reaction, the production of binding Env IgG appeared faster and durable when adjuvanting with SMNP. Strikingly, authors quantified bone marrow plasma cells and demonstrated their persistence in groups immunized with SMNP and 3M-052, indicating a robust and long-lasting antibody response. Last, authors investigated NeutAb titers. Neutralizing assays, performed by 2 independent teams, confirmed a advantage of pSer:alum + SMNP for inducing tier-2 HIV autologous NeutAb, but unfortunately not for heterologous Tier-2 strains. Nevertheless, authors established competitive binding assays and staining of Env memory B cells using modified trimer, and obtained thus evidence that pSer:alum + SMNP reduces undesirable base-binding Env-specific IgG and B memory cell responses.

The work is a comprehensive study, very-well writing, with rational of each experiment well given. Authors used advanced and appropriate techniques, and data have been carefully analyzed. Figures have been selected to highlight the quality of their data and all appropriate controls that are requested are presented in the supplementary figures. Methods are presented in detail. Data support conclusions of the authors, and any weakness of the study (EMPEM, NeutAb) is discussed honestly. Any complementary experiments would be out-of-the-scope of this specific study. Authors do not overestimate their conclusions.

A: Thank you for your positive assessment overall. We have also added new data, where we performed BCR sequencing on Env-binding B cells sorted from the lymph node fine needle aspirates. We assessed somatic mutation rates and isotype compositions of class-switched sequences over time. We also performed ranked clonal abundance curves post-priming immunization at week 6. With this new data, we hope to fully round out this work and to be a comprehensive resource for future studies.

Globally, results underscore the importance of potent adjuvant such as SMNP to prime GC/Tfh responses. We are aware that repeating trimeric Env might be necessary to induce potent NeutAb and therefore, the advantage of pSer:Alum + SMNP could be questionable. Authors discuss their critical findings, especially the fact that pSer:Alum + SMNP might be an advantage during the priming for limiting B cell immunodominance. This work will be of importance for future preclinical assay (eg testing mode of delivery of antigen). Vaccine tools and approaches are also relevant for the development of clinical trials.

A: We agree that pSer:alum + SMNP do not appear to offer an added advantage post-booster immunizations, but appear to elicit robust responses after a priming immunization. We have updated the Discussion to further comment on this topic:

“GCs are sites of antibody affinity maturation and SHM, necessary for most neutralizing antibody responses. Here, direct sampling of the GC responses in the draining LNs over time revealed both robust B_{GC} cell and GC-T_{FH} cell responses from animals immunized with pSer:alum + SMNP post-priming immunization. A probable explanation for the enhanced B_{GC} and Env-specific B_{GC} cell responses at the priming immunization is that a combined immunization regimen not only allowed greater B cell access to antigen, but also promoted B_{GC} cell differentiation by inducing greater numbers of GC-T_{FH} cells and higher quality of GC-T_{FH} cell support to B_{GC} cells. GC response advantages observed post-prime induced by pSer:alum + SMNP were not maintained with subsequent booster immunizations. Groups where GC and antibody responses were low in comparison to pSer:alum + SMNP at prime appear to increase and become comparable after multiple booster immunizations. Given the importance of durability in vaccine-induced immune responses, we postulate that having multiple booster immunizations – especially with potent adjuvants such as SMNP and 3M-052 – can help compensate for a less optimal priming immunization.”

However, the data indicate that booster immunizations can not fully compensate for shortcomings during priming, and thus the improved priming GCs for pSer are notable. This is a topic for clarification also raised by Reviewer 2. We added further consideration of this topic in the Discussion:

Nevertheless, the priming immunization is particularly important, as a series of studies have indicated that the priming immunization is the primary opportunity for recruiting rare precursor B cells—such as autologous nAb-capable B cells and bnAb-class precursors—and booster immunizations usually do not compensate for insufficient recruitment of rare precursors at the initial priming immunization^{12,15,24}. This idea is consistent with the enhanced induction of nAb responses for the pSer:alum + SMNP group compared to the other adjuvant conditions. Thus, the improved post-prime GC responses to pSer:alum + SMNP are notable.

Reviewer #3 (Remarks to the Author):

In this study conducted by Ivy Phung et al., the immunogenicity of five different HIV vaccine immunization types was examined, with variations in immunogen, adjuvants, and potential in-vivo antigen delivery kinetics. The researchers modified an HIV Env trimer by incorporating phosphoserine peptide linkers to improve its binding to aluminum hydroxide (pSer:alum). The study design involved a complex arrangement of 5 arms with 6 rhesus macaques per group, examining combinations of the original or modified trimers with three different adjuvants (alum, SMNP, 3M-052). According to the work, the combination of pSer:alum with the adjuvant SMNP resulted in enhanced germinal center (GC) responses and GC-TFH cell help. Notably, the adjuvants 3M-052 and SMNP, in combination with pSer:alum, led to the presence of long-lasting Env-specific bone marrow antibody-secreting cells. Furthermore, modifying the trimer with pSer reduced the targeting of non-neutralizing epitopes. While this manuscript is well-written and contributes to the exploration of different adjuvants in HIV vaccines, there are some important considerations to address. Briefly, the manuscript lacks experimental evidence to support the superiority of one approach over another and lacks of results that explain the mechanisms underlying the observed differences in immune responses. Specific comments are below.

A: Thank you for your comments and suggestions. We have also added new data, where we performed BCR sequencing on Env-binding B cells sorted from the lymph node fine needle aspirates. We assessed somatic mutation rates and isotype compositions of class-switched sequences over time. We also performed ranked clonal abundance curves post-priming immunization at week 6. With this new data, we hope to fully round out this work and to be a comprehensive resource for future studies.

1. The description of antigen delivery kinetics could be expanded, as it is currently limited to few words and a supplemental figure.

A: Thank you for your comment. The adjuvants and antigen delivery kinetics have previously been described (pSer: Moyer, et al., *Nature Medicine*, 2020. SMNP: Silva, Kato, et al., *Science Immunology* 2021. pSer:alum + SMNP: Rodrigues, et al., *Science Advances*, 2021). However, we have added more descriptions of the adjuvants and antigen delivery kinetics to further expand from these publications in the Results section:

“Five adjuvant technologies were tested: (i) aluminum hydroxide (MD39 adsorbed to alum, Alhydrogel), (ii) pSer:alum (pSer-MD39 bound to alum), (iii) pSer:alum + SMNP (pSer-MD39 bound to alum mixed with SMNP), (iv) SMNP (soluble MD39 mixed with SMNP), and (v) pSer:alum-3M-052 (pSer-MD39 trimer bound to alum bearing preadsorbed 3M-052 TLR7/8 agonist). We have previously described a novel approach to augment protein subunit vaccines by modifying the extrinsic properties of a vaccine – antigen delivery kinetics. Site-specific introduction of pSer tags onto protein immunogens allows for a tight binding of the antigen to the surface of alum via a ligand exchange reaction. This approach has been applied with various immunogens, from small protein constructs such as HIV engineered outer domain gp120 containing the CD4 binding site (eOD-GT8) and SARS-CoV-2 receptor binding domain (RBD) to larger HIV Env trimers (MD39)^{21,22}. pSer-conjugated MD39 exhibited high levels of initial binding to alum and retention on alum in the presence of mouse and RM serum, and preserved antigenicity when bound to alum (Figure S1a-c).”

We have also included extra background information where pSer:alum + SMNP were tested together in mice, using RBD as the antigen.

"Group 3 represented a first test of the adjuvanticity of a combination of pSer and SMNP technologies in a large animal model. A previous study done in mice investigating humoral responses to immunization combining pSer-RBD with an alum-binding co-adjuvant, SMNP (pSer-RBD:alum + SMNP), resulted in longer antigen retention, higher anti-RBD IgG titers, and enhanced neutralizing antibody responses over pSer-RBD:alum, indicating synergy between co-adjuvants²²."

2. The statistical justification for having only six rhesus macaques per group is missing, raising questions about the ability to detect the reported immunological differences with this sample size. In fact, both Figure 1 and Figure 2 relied on values obtained from two lymph nodes collected from the same animals to achieve statistical significance. This approach of dividing the vaccine dose and administering it bilaterally is not translatable to humans, and the use of values from the same animal without proper identification might be improper. Are there any differences between values obtained from two lymph nodes in the same animal? If yes, why? Why did researchers not plot the average of the results obtained from the two lymph nodes if the study was properly powered? The utilization of values obtained from two lymph nodes collected from the same animals to achieve statistical significance is questionable and may not accurately reflect biological duplicates. It would be beneficial to provide clarification on any potential differences between the values obtained from the two lymph nodes and consider plotting the average results or to show matching results.

A: Thank you for your comments. We agree that statistical justification for animal sample size is certainly important. As in all NHP studies, a balance between animal numbers and costs must be considered. We have previously performed a power analysis for our NHP experiments that determined that with a sample size of 6 animals, we can distinguish a 2 to 3-fold difference in mean nAb titers with 90% confidence. That calculation was successful in Pauthner et al. 2017, and we have used it in several of our NHP studies since. Additionally, preliminary data allowed us to also perform power calculations for assessing Env-specific B_{GC} increases indicating the current study was sufficiently powered. This has been reflected in the text in the Methods under the Animals and immunizations section:

"Animals were grouped to match age, weight, and sex. All animals were between 3.5-5 years old at the time of the priming immunization, with each study group consisting of 3 females and 3 males (n=6/group). A power analysis was previously performed to determine the optimal sample size to distinguish meaningful statistical differences in nAb titers between animal groups²⁴."

Regarding bilateral immunizations and interpretations, we apologize for being unclear. This is an important topic on which we have done substantial previous experiments. We posited that B_{GC} cell and GC-T_{FH} cell reactions in draining LNs from separate limbs to be largely independent after priming immunizations, and that we would take advantage of that biology to double the number of samples available for GC analyses. We first assessed this in Havenar-Daughton, et al., *Cell Reports*, 2016, and then again in Pauthner, et al., *Immunity*, 2017. In those studies, we indeed observed independent GC responses occurring in the left and right draining LNs of the same animal. We then formally tested immunogen drainage, wherein animals immunized unilaterally did not have measurable GC responses above baseline levels in the contralateral LN (Havenar-Daughton, et al., *Cell Reports*, 2019). In light of those numerous observations, we have continued to make use of the bilateral immunization and LN FNA GC analyses approaches in multiple successful studies (Cirelli, et al., *Cell*, 2019; Silva, Kato, et al., *Science Immunology*, 2021). This has been reflected in the text in the Results section for further clarification:

"All animals were immunized bilaterally in the left and right thigh. As germinal center (GC) responses in contralateral limbs were found to be largely independent after a priming immunization^{15,24,25}, a bilateral immunization strategy increases the number of lymph nodes (LNs) available for sampling GCs."

3. The claim of superiority of one approach over another lacks support from experiments that could explain the mechanisms behind the observed "larger," "faster," and "more durable" immune responses. The designation of a faster antibody response based on a single measurement at week 2 seems arbitrary. Moreover, at week 4, when the antibody peak is reached and an objective threshold can be identified, no differences were observed between the pSer:alum + SMNP group and others, raising questions about the necessity of trimer modification and alum to achieve similar outcomes in clinical studies. The lack of significant differences in antibody quantity at week 10, despite differences in B- and T-cells at the same time point, further complicates interpretation.

A: Thank you for your comment. We highlighted week 2 as it is rare to observe Env-binding IgG responses elicited by a conventional alum immunization at this timepoint. Therefore, we thought it was an interesting observation to be able to detect a large difference in Env-binding IgG titers from pSer:alum + SMNP compared to all other groups. However, the total Env-binding IgG titers do not tell the full story and are not a main conclusion of the study. The neutralizing antibody response, the magnitude of the BM plasma cells, and the magnitude and durability of the GC responses are major findings. At week 10, there is no significance difference in the total Env-binding IgG titers between pSer:alum and alum. However, the 19R base-binding ELISAs show a difference in Env-binding specificities. While the total IgG binding titers were comparable between these two groups, alum elicited antibodies that bound more to the off-target base of the trimer while pSer:alum elicited more on-target antibodies. Similar trends are also seen between pSer:alum + SMNP and pSer:alum-3M-052 compared to SMNP. We do not expect the GC responses and antibody responses to correlate at the same timepoint. Often, there is a delay in the kinetics between the GCs and outcomes measured in peripheral blood. While total Env-binding IgG titers may be comparable between groups at week 10, animals that received pSer-trimer modification trended higher in their non-base-binding Env specificities. Those findings were consistent with the finding of more robust neutralizing antibody titers after pSer:alum + SMNP compared to pSer:alum or alum alone, match the differences in the germinal center responses between those groups.

Related to this is the topic of immunodominance and the importance of the priming immunization. In response to requests by reviewers 1 and 2 we have updated the Discussion to further comment on this topic:

"GCs are sites of antibody affinity maturation and SHM, necessary for most neutralizing antibody responses. Here, direct sampling of the GC responses in the draining LNs over time revealed both robust B_{GC} cell and GC-T_{FH} cell responses from animals immunized with pSer:alum + SMNP post-priming immunization. A probable explanation for the enhanced B_{GC} and Env-specific B_{GC} cell responses at the priming immunization is that a combined immunization regimen not only allowed greater B cell access to antigen, but also promoted B_{GC} cell differentiation by inducing greater numbers of GC-T_{FH} cells and higher quality of GC-T_{FH} cell support to B_{GC} cells. GC response advantages observed post-prime induced by pSer:alum + SMNP were not maintained with subsequent booster immunizations. Groups where GC and antibody responses were low in comparison to pSer:alum + SMNP at prime appear to increase and become comparable after multiple booster immunizations. Given the importance of durability in vaccine-induced immune responses, we postulate that having

multiple booster immunizations – especially with potent adjuvants such as SMNP and 3M-052 – can help compensate for a less optimal priming immunization.”

However, the data indicate that booster immunizations cannot fully compensate for shortcomings during priming, and thus the improved priming GCs for pSer are notable. This is a topic for clarification also raised by Reviewer 2. We added further consideration of this topic in the Discussion:

Nevertheless, the priming immunization is particularly important, as a series of studies have indicated that the priming immunization is the primary opportunity for recruiting rare precursor B cells—such as autologous nAb-capable B cells and bnAb-class precursors—and booster immunizations usually do not compensate for insufficient recruitment of rare precursors at the initial priming immunization^{12,15,24}. This idea is consistent with the enhanced induction of nAb responses for the pSer:alum + SMNP group compared to the other adjuvant conditions. Thus, the improved post-prime GC responses to pSer:alum + SMNP are notable.

4/5. It is important to note that the efficacy of these vaccines has not been demonstrated through mucosal viral challenges, which is a significant lack for readers of this journal. Indeed higher immunogenicity is not often translated in higher vaccine efficacy or vaccine efficacy at all in the context of strategies to prevent HIV. Even accepting the lack of mucosal viral exposures and considering this as an immunogenic under-powered study it is a missed opportunity the lack of data of mucosal sites, where viral transmission happens.

A: This study was designed to study the immunogenicity of the different adjuvant technologies and not as a challenge study. We apologize for any confusion. We recognize the importance of vaccine-induced protection against mucosal viral challenges, and we have incorporated that into our overall program, which is one critical feature of the current study. We have previously shown that Tier 2 nAb titers against BG505 after BG505-base Env trimer immunizations were a clear correlate of protection in NHPs against a mucosal viral challenge. When we immunized NHPs with BG505, animals that produced an autologous serum ID₅₀ nAb titer of approximately 1:500 afforded ~90% protection from a medium-dose rectal SHIV_{BG505} infection (Pauthner, et. al., *Immunity*, 2019). Thus, in this study we have maintained usage of BG505-based Env trimers so as to relate our immunogenicity results to known SHIV mucosal challenge vaccine outcomes. For this study, at the peak of the nAb responses at week 42, 4 out of 6 animals immunized with pSer:alum + SMNP developed nAbs with an ID₅₀ of 1:500 or higher, in contrast to alum or pSer:alum groups with 1 out of 6 animals in each group. Given the data, based on the nAb titers elicited, we would anticipate better protection for animals immunized with pSer:alum plus SMNP against tier 2 SHIV_{BG505} pseudovirus infection. This is now reflected in the Discussion:

“Vaccine-induced protection from mucosal viral exposures is an important end goal in the development of an antibody-based HIV vaccine. Previously, it was found that nAb titers were a correlation of protection against SHIV_{BG505} viral challenge in NHPs. Animals that were immunized with BG505 Env trimer and produced an autologous serum ID₅₀ nAb titer of 1:500 or better were afforded ~90% protection from a medium-dose homologous SHIV_{BG505} infection³⁵. For this study, 4 out of 6 animals immunized with pSer:alum + SMNP developed peak nAbs with an ID₅₀ of 1:500 or better, in contrast to alum or pSer:alum groups with 1 out of 6 animals in each group. Given the data, based on the nAb titers elicited, we would anticipate better protection against tier-2 SHIV_{BG505} pseudovirus infection for animals immunized with BG505 Env trimer plus pSer:alum and SMNP compared to BG505 Env trimer plus alum.”

Overall, while the manuscript is well-written, the study is highly descriptive with flaws in its design. We encourage further work to solidify these preliminary findings.

A: We appreciate the reviewer's desire for additional data. We have addressed specific criticisms above, including the design of the study, the methodology, the relevance for protective immunity against mucosal challenge, and the relevance of vaccine immune response differences after the priming immunization.

Reviewers' Comments:

Reviewer #3:

Remarks to the Author:

We thank the authors for addressing several of our observations. The description of antigen delivery kinetic has been expanded. The issue about power calculation has been addressed by referring to previous work. Data visualization/plotting in main figures has not been addressed. In fact, my understanding is correct, values obtained from two separate anatomical sites from the same animal have been plotted in several instances (Fig 1d, 1f, 1g, Fig2c, 2d, 2e) with two separate but indistinguishable markers on the same graph so that it is not possible to identify the values from the same animals and distinguish left/right sites. Understanding the variance in results from the same animal is crucial for interpreting the data (or confirming previous published work – as mentioned by the authors) and its implications more accurately. In particular, the authors refer to past work that demonstrated that independent GC responses occur in the left and right draining LNs of the same animal and that germinal center (GC) responses in contralateral limbs were found to be largely independent after a priming immunization. This explanation reinforces why it is important to present with clear distinguishable markers on the graphs lymph nodes from the same animals and left/right side. This will enhance the clarity and precision of the study but also facilitate a more accurate interpretation of results by peers (maybe adding a supplemental figure comparing left/right side data?). The absence of efficacy data has been kindly indirectly introduced in the discussion and serves to underscore that the conclusions about efficacy are based largely on findings from previous studies. While promising indications have been observed for animals immunized with BG505 Env trimer plus pSer:alum and SMNP compared to BG505 Env trimer plus alum, further validation against tier-2 SHIVBG505 pseudovirus infection remains a valuable area of exploration and additional outcomes merit consideration

Reviewer #3 (Remarks to the Author):

We thank the authors for addressing several of our observations. The description of antigen delivery kinetic has been expanded. The issue about power calculation has been addressed by referring to previous work. Data visualization/plotting in main figures has not been addressed. In fact, my understanding is correct, values obtained from two separate anatomical sites from the same animal have been plotted in several instances (Fig 1d, 1f, 1g, Fig2c, 2d, 2e) with two separate but indistinguishable markers on the same graph so that it is not possible to identify the values from the same animals and distinguish left/right sites. Understanding the variance in results from the same animal is crucial for interpreting the data (or confirming previous published work - as mentioned by the authors) and its implications more accurately. In particular, the authors refer to past work that demonstrated that independent GC responses occur in the left and right draining LNs of the same animal and that germinal center (GC) responses in contralateral limbs were found to be largely independent after a priming immunization. This explanation reinforces why it is important to present with clear distinguishable markers on the graphs lymph nodes from the same animals and left/right side. This will enhance the clarity and precision of the study but also facilitate a more accurate interpretation of results by peers (maybe adding a supplemental figure comparing left/right side data?). The absence of efficacy data has been kindly indirectly introduced in the discussion and serves to underscore that the conclusions about efficacy are based largely on findings from previous studies. While promising indications have been observed for animals immunized with BG505 Env trimer plus pSer:alum and SMNP compared to BG505 Env trimer plus alum, further validation against tier-2 SHIVBG505 pseudovirus infection remains a valuable area of exploration and additional outcomes merit consideration.

A: Thank you for your comment. We have added new figure panels in the Supplementary Figures denoting the left and right lymph node (LN) for multiple GC measurements at the week 3 timepoint: B_{GC} cells (% B cells) (**Supplementary Fig. 2b**), Env⁺ B_{GC} cells (% B cells) (**Supplementary Fig. 2g**) and GC-T_{FH} cells (% CD4⁺ T cells) (**Supplementary Fig. 3a**). The left and right lymph LN are denoted by either a filled or an empty circle, with each unique color denoting a different animal within the immunization group. Matched colors indicate left and right LNs of the same animal. The B_{GC} and GC-T_{FH} cell frequencies indicate that the left and right draining LNs are largely independent in our adjuvant study, corroborating previous studies (Havenar-Daughton, et al., *Cell Reports*, 2016; Pauthner, et al., *Immunity*, 2017; Havenar-Daughton, et al., *Cell Reports*, 2019). We have updated the text to refer to these specific figure panels when discussing the post-prime responses:

"After a single priming immunization, pSer:alum + SMNP induced higher total GC B cell (B_{GC}, CD38⁺CD71⁺) frequencies than all other immunization groups, peaking at week 3 (**Fig. 1b-d, Supplementary Fig. 2a-b**).

Env-binding B_{GC} cells (CD38⁺CD71⁺/Env⁺, **Fig. 1b, 1e-f, Supplementary Fig. 2f-i**) and total Env-binding B cells (CD3⁺CD20⁺/Env⁺, **Supplementary Fig. 2j-m**) were quantified.

A single bolus priming immunization of pSer:alum + SMNP with Env trimer elicited the largest GC-T_{FH} cell response (PD-1^{hi}CXCR5⁺ of CD4⁺CD8a⁺) of all immunization groups, peaking at week 3 (**Fig. 2a-c, Supplementary Fig. 2a, Supplementary Fig. 3a**)."

The editor also requested updates to the figures and text to conform to Nature style guidelines in the Nature checklist, and those have now been completed.